# Intermittent scavenging of storage lesion from stored red blood cells by electrospun nanofibrous sheets enhances their quality and shelf-life

**Subhashini Pandey** [1,2,3], **Manohar Mahato** [1,3], **Preethem Srinath** [1], **Utkarsh Bhutani** [1], **Tanu Jain Goap** [1,2], **Priusha Ravipati** [1] & **Praveen Kumar Vemula** [1] ✉

Transfusion of healthy red blood cells (RBCs) is a lifesaving process. However, upon storing RBCs, a wide range of damage-associate molecular patterns (DAMPs), such as cell-free DNA, nucleosomes, free-hemoglobin, and poly-unsaturated-fatty-acids are generated. DAMPs can further damage RBCs; thus, the quality of stored RBCs declines during the storage and limits their shelf-life. Since these DAMPs consist of either positive or negative charged species, we developed taurine and acridine containing electrospun-nanofibrous-sheets (*Tau-Acr*NFS), featuring anionic, cationic charges and an DNA intercalating group on their surfaces. We show that *Tau-Acr*NFS are efficient in scavenging DAMPs from stored human and mice RBCs ex vivo. We find that intermittent scavenging of DAMPs by *Tau-Acr*NFS during the storage reduces the loss of RBC membrane integrity and reduces discocytes-*to*-spheroechinocytes transformation in stored-old-RBCs. We perform RBC-transfusion studies in mice to reveal that intermittent removal of DAMPs enhances the quality of stored-old-RBCs equivalent to freshly collected RBCs, and increases their shelf-life by ~22%. Such prophylactic technology may lead to the development of novel blood bags or medical device, and may therefore impact healthcare by reducing transfusion-related adverse effects.

Blood transfusion is often a lifesaving practice for patients in the intensive care unit (ICU), where ~50–70% of ICU patients are transfused with blood units during their stay[1]. Typical indications for blood transfusion include sickle cell crisis, anemia, and severe blood loss[2]. The highest transfused blood components are red blood cells (RBCs), with >85 million RBC units are transfused annually worldwide[2,3]. While RBC transfusion has therapeutic benefits, transfusion of storage-aged RBCs may cause deleterious effects on the recipients. These deleterious effects have been attributed to the storage lesion consisting of damage-associated molecular patterns (DAMPs) generated due to biochemical, morphological, and structural changes in RBCs during the storage-aging process[4,5]. The most studied DAMPs in this context are as follows: (i) extracellular free-iron and free-hemoglobin (Hb) generated due to lysis of RBCs, (ii) bioactive lipids such as poly-unsaturated fatty acids (PUFAs), (iii) extracellular DNA, and (iv) nucleosomes generated by neutrophils. These DAMPs are a potential

[1]Institute for Stem Cell Science and Regenerative Medicine (inStem), GKVK Post, Bellary Road, Bangalore 560065 Karnataka, India. [2]The University of Trans-Disciplinary Health Sciences and Technology, Attur (post), Yelahanka, Bangalore 560064 Karnataka, India. [3]These authors contributed equally: Subhashini Pandey, Manohar Mahato. ✉e-mail: praveenv@instem.res.in

source of sequelae in vulnerable patients[6]. Based on storage-time-dependent accumulation of DAMPs, stored human RBC units are classified as young RBCs or fresh RBCs (RBCs of <14–21 days of storage) and old RBCs (stored between 21 and 42 days)[7]. Storage of RBCs induces a sequence of biochemical and biomechanical changes in a progressive manner that affect deformability, cell viability, micro-circulatory flow, and oxygen-carrying capability of RBCs[8]; hence, the detrimental effect of RBC storage can negatively impact the transfusion outcomes. Therefore, developing strategies and technologies to enhance the quality and shelf-life of stored RBCs to reduce the incidence of transfusion-related complications will significantly impact healthcare.

The United States Foods and Drug Administration (US-FDA) regulation states that human-packed RBCs (pRBCs) can be stored up to 42 days before transfusion. Post 42 days, the increased DAMPs (extracellular DNA, nucleosomes, Hb, and bioactive lipids) pose a fatal risk to the patient by eliciting an immunomodulatory response[9,10]. Leukocyte-associated DAMPs, such as extracellular DNA and nucleosomes, are potential immune elicitors in recipients and are implicated in transfusion-related complications[11]. Leukodepletion of packed RBCs using specialized filters mitigates risk to a large extent, putatively by reducing a load of extracellular DNA and nucleosomes in blood units[12]. However, leukodepletion does not remove other critical RBCs-generated DAMPs, such as free-Hb and bioactive lipids, which are also significant mediators of transfusion-related complications. Therefore, in alternative to the leukoreduction process, studies thus far have focused on improving storage conditions by alternative cryopreservation protocols[13,14], anaerobic storage[15,16], and usage of additives/rejuvenation solutions[17–19]. Additionally, washing stored RBCs with saline (0.9% NaCl) to remove accumulated bioactive factors is another established strategy approved to use in transfusion medicine[20,21]. Past investigations suggest that adding preservatives, storing RBCs in an alkaline hypotonic solution with antioxidants, or rendering anaerobic storage conditions improve storage time by up to 10–15 days[17–19]. Although these approaches have shown encouraging results, efficient technologies to scavenge the spectrum of DAMPs and further reduce their formation in stored RBCs remain an unmet need. Additionally, the leukoreduction process is expensive. Hence, most blood banks in India and developing countries do not routinely use leukoreduction processes. Interestingly, prior to initiating this work, our discussions with blood banks in India revealed that only <10% of blood banks use the leukoreduction process. It is used only in cases where recipients are immunosuppressed/immuno-compromised. Therefore, an important driver for developing new technology is the desire to develop an affordable intervention that could be used in developing and under-resourced countries where non-leukoreduced blood products are the standard inventory.

Here we develop a new approach, demonstrating that intermittent scavenging of DAMPs using electrospun-charged nanofibrous sheets slows the deterioration of stored RBCs, increasing their quality and shelf-life. Critical DAMPs such as extracellular DNA, nucleosomes, free-Hb, and PUFAs are charged molecules with anionic or cationic nature. Therefore, we hypothesized that *charged nanofibrous sheets* made with cationic and anionic polymers may scavenge DAMPs through ionic interactions (Fig. 1a). We have synthesized novel polymers with anionic and cationic nature by introducing poly taurine and poly acridine, respectively. An efficient process to fabricate electrospun nanofibrous sheets with anionic (*Tau*-NFS) and cationic (*Acr*-NFS) charges has been generated. Additionally, the acridine group in *Acr*-NFS may act as a DNA intercalator. We have demonstrated that a combination of *Tau*-NFS and *Acr*-NFS, hereafter *Tau-Acr*NFS, is efficient in scavenging storage lesions such as free extracellular DNA, nucleosomes, Hb, and PUFAs from stored human and mice RBC units, ex vivo. Two major phenomena were demonstrated using *Tau-Acr*NFS;

i) after storage lesion formed in human old RBCs (stored for 42 days), *Tau-Acr*NFS could efficiently scavenge and remove accumulated DAMPs, and ii) the intermittent scavenging of DAMPs using *Tau-Acr*NFS either at 21st or 28th day resulted in significantly less production of DAMPs at maximum storage time (on 42nd day), and reduced the loss of RBC membrane integrity. We have demonstrated that intermittent removal of DAMPs either on 21st or 28th day by *Tau-Acr*NFS enhanced the quality of the blood and increased the shelf-life by ~22%. Additionally, the RBC transfusion studies suggested that the quality of intermittently treated and stored for a maximum allowed 14 days (old RBCs) is equivalent to the quality of freshly collected RBCs without storage. These results are remarkable, and we aim to generate either novel blood bags or an insert-based medical device using *Tau-Acr*NFS, which may have an enormous impact on improving the quality and shelf-life of stored blood.

## Results
### Charged electrospun nanofibrous sheets efficiently scavenge DAMPs
To synthesize polymers with anionic/cationic charges to generate electrospun nanofibrous sheets, we have selected poly(methyl vinyl ether-*alt*-maleic acid) [PMVEMA, average Mw: 330,000], to which 50% of tetradecylamine was functionalized to impart hydrophobic nature to the polymer. Subsequently, either 25% of taurine or 11% of hexyl acridine were functionalized to obtain anionic (poly-Tau) and cationic (poly-Acr) polymers, respectively. In addition to furnishing cationic charge, acridine can also bind DNA by intercalation, which could enhance the ability of *Acr*-NFS to scavenge extracellular DNA. The detailed synthesis scheme is provided in Supplementary Fig. 1. Anionic (*Tau*-NFS) and cationic (*Acr*-NFS) electrospun nanofibrous sheets were generated through conventional electrospinning of *poly*-Taurine and *poly*-Acridine, respectively (Methods). *Tau*-NFS and *Acr*-NFS are comprised of nanofibers with 100–200 nm width and >50–100 micron length (Fig. 1b). Surface charge measurements confirmed the overall negative and positive charge of *Tau*-NFS and *Acr*-NFS, respectively (Fig. 1c). Furthermore, contact angle measurements show that both NFS are hydrophobic (Fig. 1d), which is critical to prevent aqueous fluid absorption. A 2-h incubation of *Tau*-NFS and *Acr*-NFS in optisol (AS-5) solution that is used as a preservative for storing RBCs suggested that NFS are stable in preservative solution, and the quantification of leached polymers shows that neither polymers have any significant leaching (<0.2%). Scanning electron microscope images of *Tau*-NFS and *Acr*-NFS post-incubation with stored RBCs show that RBCs or neutrophils do not adhere to NFS (Fig. 1e); hence, there will not be any loss of RBC count due to the NFS treatment. Furthermore, to understand whether treatment of *Tau*-NFS and *Acr*-NFS can cause cell loss, we performed a complete blood count (CBC) before and after treatment with *Tau*-NFS and *Acr*-NFS. Quantifying the number of RBCs, white blood cells, platelets, neutrophils and lymphocytes before and after treating with NFS (Supplementary Fig. S2) revealed that there is no loss of cells suggesting that these nanofibrous sheets do not cause any cell death or cell capture. Additionally, the hemolysis rate was measured upon incubation of RBCs with *Tau-Acr*NFS at 37 °C for 1 h. A <1% of hemolysis was observed for *Tau-Acr*-NFS (Fig. 1f). Cumulatively, these results suggest that *Tau-Acr*NFS are stable and do not cause significant hemolysis. However, it should be noted that to evaluate hemocompatibility quantitatively, highly sensitive chemical analyses, such as by untargeted liquid chromatography–mass spectrometry (LC–MS/MS), will be required to determine any biochemical changes following exposure of RBCs to *Tau-Acr*NFS treatment. These experiments will be done in the next study.

Before testing the DAMPs-scavenging efficacy of NFS, we quantified the production of DAMPs (extracellular DNA, Hb, and PUFAs) in stored human non-leukoreduced RBCs in a time-dependent manner for up to 42 days. Various analytical techniques such as PicoGreen

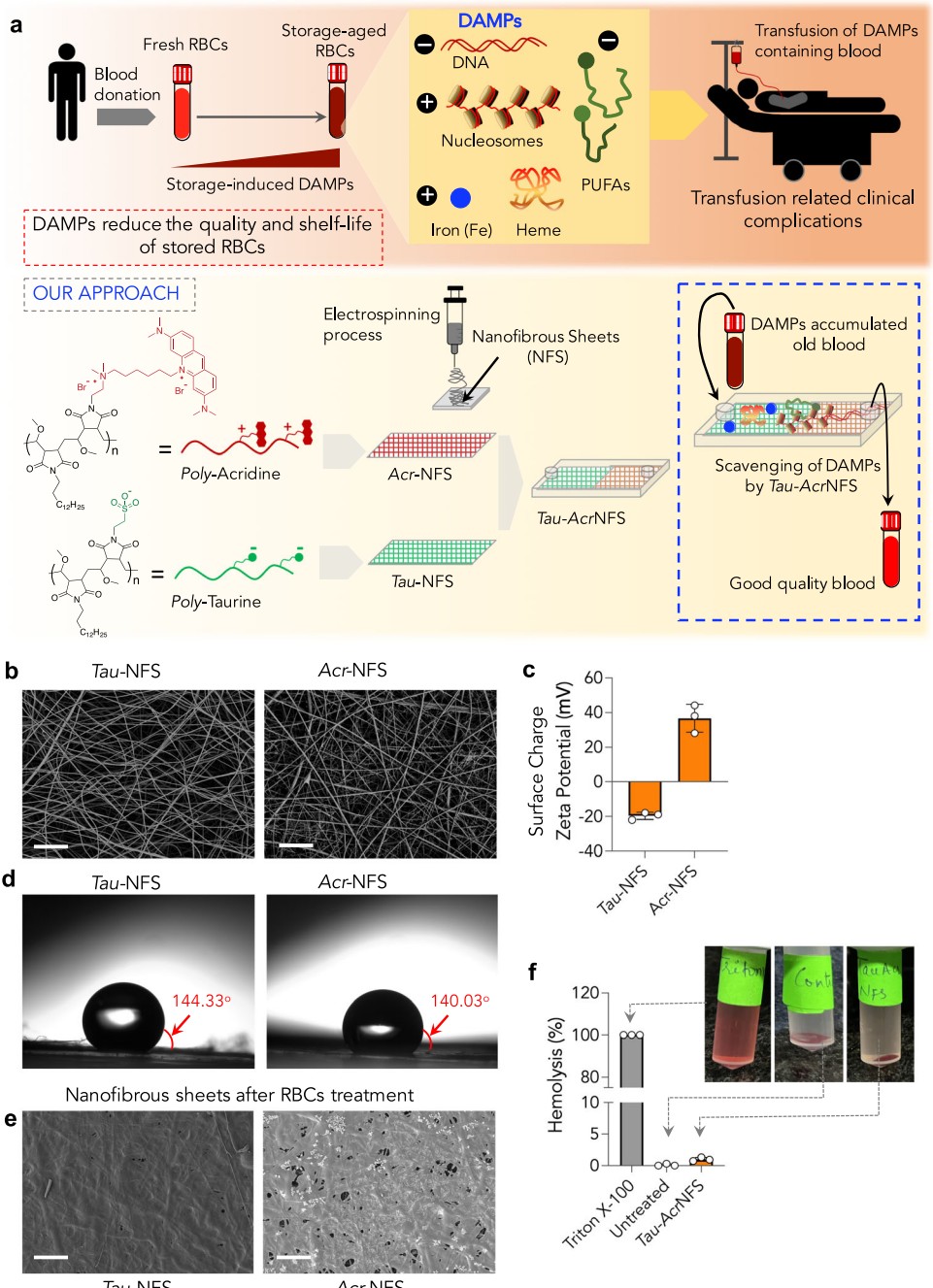

**Fig. 1 | Hemocompatible *charged nanofibrous sheets* scavenge storage lesions to improve the quality of blood. a** Stored RBCs produce storage lesion or damage-associated molecular patterns (DAMPs) such as DNA, nucleosomes, heme (Hb), iron, and polyunsaturated fatty acids (PUFAs). Transfusion of DAMPs containing RBCs could lead to transfusion-related complications, including systemic inflammation and organ injury. The presence of DAMPs progressively reduces the quality of stored RBCs and limits their shelf-life. All DAMPs consist of positive or negative charge entities. Therefore, *in our approach*, we designed hemocompatible polymers poly-acridine and poly-taurine composed of complementary cationic and anionic charges, respectively. Using these polymers, charge-bearing electrospun nanofibrous sheets (*Tau*-NFS and *Acr*-NFS) were prepared to scavenge charged DAMPs from stored RBCs, which remarkably enhanced the quality and shelf-life of stored RBCs. **b** Scanning electron microscopy images of *Tau*-NFS and *Acr*-NFS reveal the presence of nanofibrous mesh with a high aspect ratio (scale bar = 2 μm, representative images were selected from 15 images from three independent experiments). **c** Surface charge of *Tau*-NFS and *Acr*-NFS indicate that *Tau*-NFS has a net negative charge (−19.7 mV) and *Acr*-NFS has a net positive charge (+36.7 mV). **d** Contact angle measurements showed that surfaces of *Tau*-NFS and *Acr*-NFS are hydrophobic, which could reduce the absorption of aqueous components during the treatment of stored blood. **e** SEM images of *Tau*-NFS and *Acr*-NFS post-treatment with stored RBCs reveal that except for deposition of DAMPs, no cells have adhered to the NFS (scale bar = 10 μm, representative images were selected from 15 images from three independent experiments). **f** Hemolysis assay revealed that <1% hemolysis was observed upon treatment with *Tau*-*Acr*NFS. Data are mean ± s.d. (*n* = 3, from independent experiments). Source data are provided as a Source Data file.

assay, ELISA, Drabkin's assay, and LC−MS were used to quantify extracellular DNA, nucleosomes, Hb, and PUFAs, respectively (Methods). Quantitative elucidation of the storage lesion revealed that DAMPs, including extracellular DNA (Fig. 2a), Hb (Fig. 2b), and

PUFAs such as arachidonic acid (AA, Fig. 2c) and hydroxyeicosatetraenoic acids (5-HETE, 12-HETE, and 15-HETE, Fig. 2d−f) increase over time. For example, on the 42nd day, extracellular DNA and Hb concentration increased by ~150-fold and 40-fold,

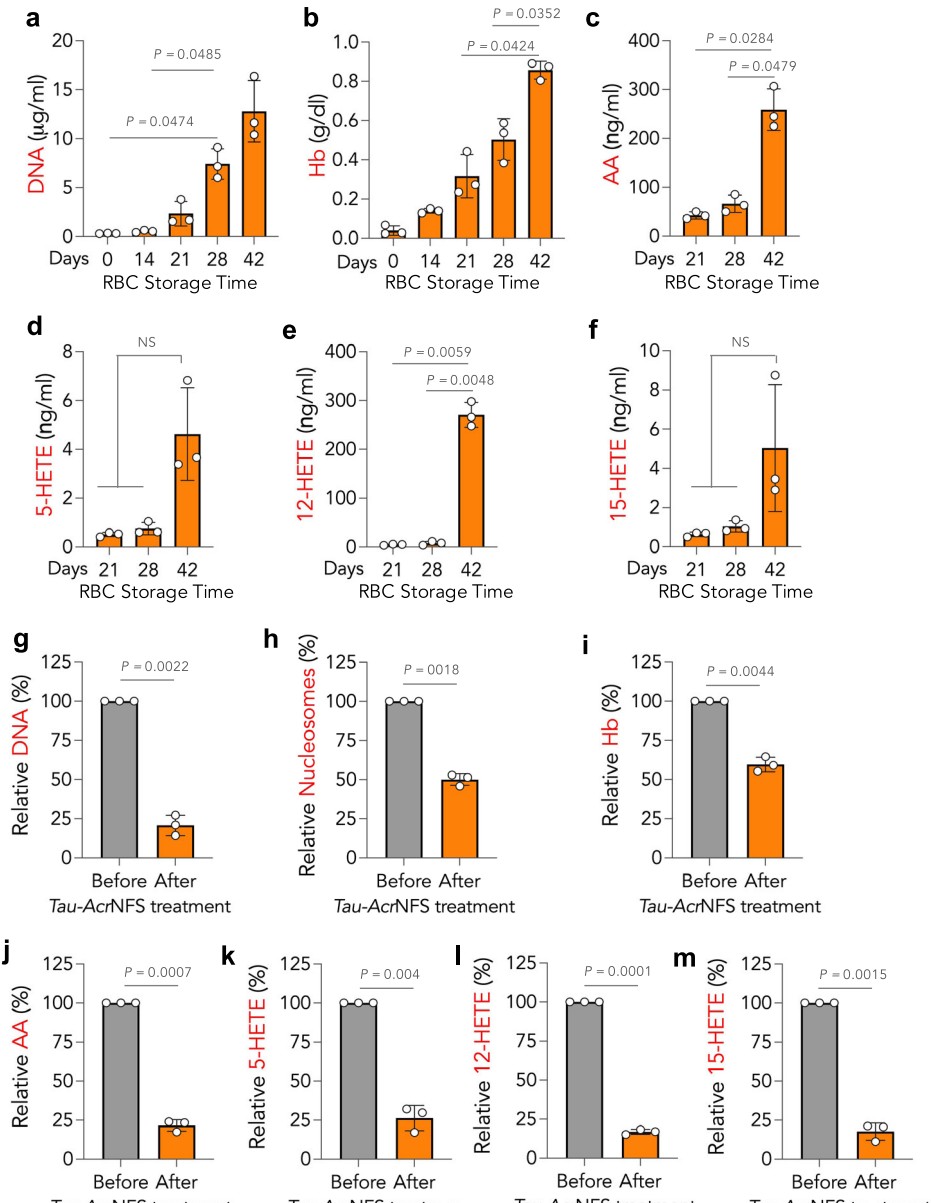

**Fig. 2 | Hypothermal storage of human RBCs induced DAMPs production and charged *Tau-Acr*NFS efficiently scavenged DAMPs from stored RBCs.**
**a–f** Storage-induced production of DAMPs as a function of time. RBCs stored at 4 °C progressively produce DNA (**a**), free hemoglobin (Hb, **b**), and polyunsaturated fatty acids (PUFAs) such as AA (**c**), 5-HETE (**d**), 12-HETE (**e**), and 15-HETE (**f**).
**g–m**, Incubation of DAMPs accumulated 42 days-stored RBCs with anionic and cationic NFS, *Tau-Acr*NFS, for 5 min at 4 °C efficiently scavenged and reduced the concentration of DAMPs significantly. The relative concentration of DAMPs before and after *Tau-Acr*NFS treatment has significantly reduced accumulated DNA (**g**), nucleosomes (**h**), Hb (**i**), and PUFAs (**j–m**). Data are mean ± s.d. (*n* = 3, from independent experiments). For **a–f**, *P* values were determined by repeated measures one-way ANOVA, and for (**g–m**), by two-tailed Student's *t*-test with Welch's correction using GraphPad PRISM 9, and exact *P* values are indicated. NS = not significant. Source data are provided as a Source Data file.

respectively, compared to the 0th day. These observations are in agreement with previous reports[22–25].

At the onset, we have systematically optimized the required surface area of nanofibrous sheets and incubation time to scavenge DAMPs. One milliliter of RBCs (42 days stored) were added to the different surface areas of sheets (8, 16, and 24 cm²) and incubated for 30 min at 4 °C. Quantification of extracellular DNA suggested that both nanofibrous sheets could scavenge DNA efficiently with an 8 cm² sheet, and above 8 cm² up to 24 cm² did not increase the scavenging capacity (Supplementary Fig. 3a). Subsequently, RBCs were incubated on 8 cm² nanofibrous sheets for 5, 15, and 30 min to identify the optimum incubation time. Data suggests that *Tau*-NFS and *Acr*-NFS could scavenge the DNA within 5 min of incubation, and

more prolonged incubation did not enhance scavenging capacity (Supplementary Fig. 3b).

To scavenge all DAMPs (extracellular DNA, nucleosomes, Hb, and PUFAs) from old RBCs, 1 ml of 42 days-stored RBCs were incubated with cationic/anionic nanofibrous sheets (*Tau-Acr*NFS) for 5 min at 4 °C. Data in Fig. 2g–m suggests that *Tau-Acr*NFS are efficient in scavenging DAMPs. Upon incubation, nanofibrous sheets scavenged 80% of DNA (Fig. 2g), 50% of nucleosomes (Fig. 2h), 45% of Hb (Fig. 2i), and 75–80% of PUFAs including AA, 5-HETE, 12-HETE, and 15-HETE (Fig. 2j–m). These results suggest that *Tau-Acr*NFS can efficiently scavenge and significantly reduce the concentrations of DAMPs in 42 days of stored old RBCs. SEM images of *Tau*-NFS and *Acr*-NFS post-treatment with stored RBCs (Fig. 1e) shows a gummy layer on the

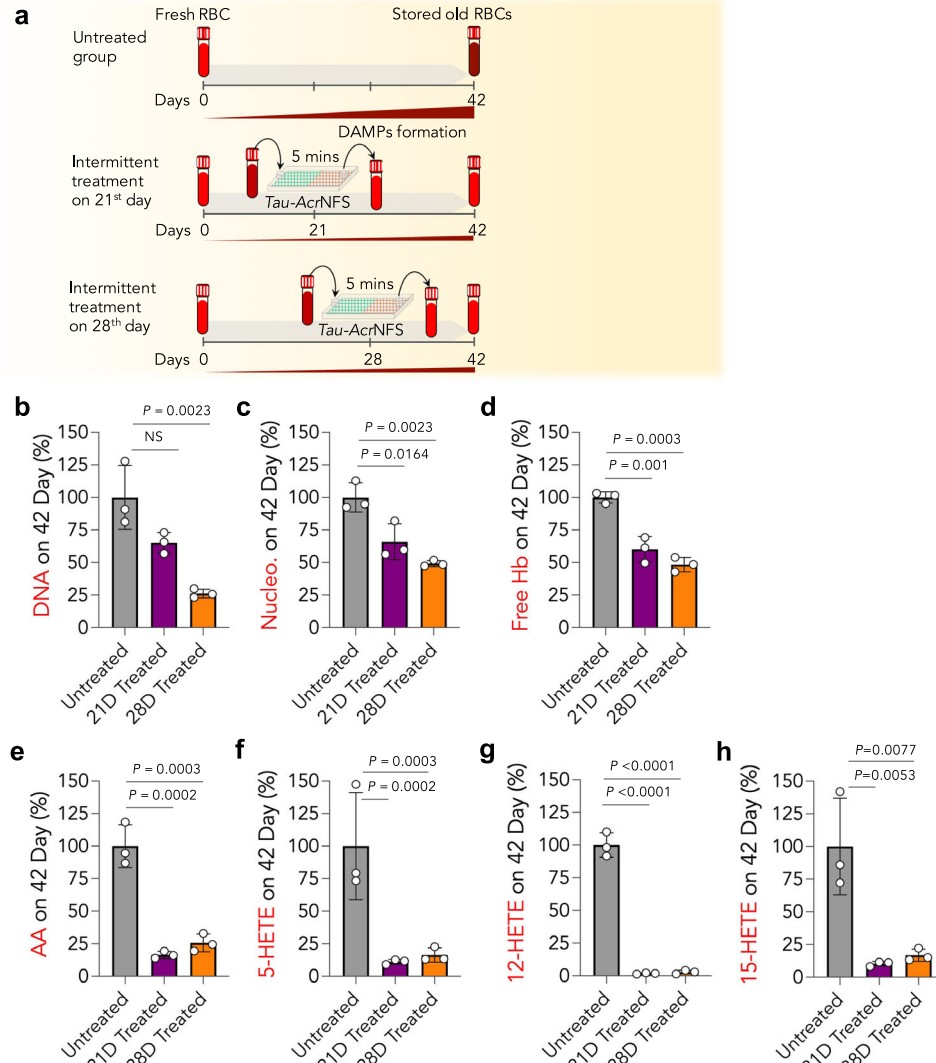

**Fig. 3 | Intermittent scavenging of DAMPs using *Tau-Acr*NFS from stored RBCs can reduce the accumulation of DAMPs at 42 days of stored human RBCs.** **a** Schematic of DAMPs production in untreated and *Tau-Acr*NFS treated stored RBCs. In the untreated group, RBCs were stored at 4 °C for 42 days, in treated groups, RBCs were incubated with *Tau-Acr*NFS for 5 min on either the 21st or 28th day, and stored at 4 °C for 42 days, and DAMPs were quantified on 42nd day for all groups (**b**–**h**). Intermittent treatment of stored RBCs with *Tau-Acr*NFS on either the 21st or 28th day significantly reduced the accumulation of DNA (**b**), nucleosomes (**c**), Hb (**d**), and PUFAs (**e**–**h**) on the 42nd day. Data are mean ± s.d. ($n = 3$, from independent experiments); $P$ values were determined by ordinary one-way ANOVA with Tukey's post hoc analysis using GraphPad PRISM 9, and exact $P$ values are indicated. Source data are provided as a Source Data file.

surface, which could be due to the accumulation of sequestered proteins and lipids on the nanofibrous sheets.

## Intermittent scavenging of DAMPs by *Tau-Acr*NFS significantly enhances the quality of stored old RBCs

The data in Fig. 2a–f suggest that DAMPs formation is minimal until 28 days of storage; subsequently, an exponential production of DAMPs occurs. It is reasoned that the accumulated DAMPs, until 28 days, might have a synergistic effect on accelerating the deterioration of stored human RBCs; hence, large quantities of DAMPs are produced within 42 days. Therefore, we envisaged that intermittent removal of DAMPs either on the 21st or 28th-day using *Tau-Acr*NFS would reduce further damage to RBCs and enhance the quality of 42 days-stored RBCs. To test this hypothesis, either on the 21st or 28th day, stored RBCs were incubated with *Tau-Acr*NFS for 5 min (Fig. 3a), stored back at 4 °C, and measured the DAMPs on the 42nd day. The data in Fig. 3b–h suggest that intermittent removal of DAMPs either on the 21st or 28th day significantly reduces the number of DAMPs present on the 42nd day. Furthermore, intermittent treatment either on the 21st or 28th day has

remarkably reduced the formation of cell-free DNA, nucleosomes, cell-free Hb, and PUFAs by 75, 50, 50, and 75–98%, respectively, compared to 42 days of stored untreated RBCs (Fig. 3b–h).

RBCs' shape and membrane rigidity play critical roles in their quality, function, and fate, such as gas transport capacity and RBC clearance from circulation[26,27]. RBCs should stretch/elongate and undergo deformation to pass through capillaries. Hence to retain function, they should be mechanically stable and withstand extensive passive deformation[28]. DAMPs are known to have a deleterious effect on RBCs' quality. The morphology of RBCs progressively deteriorates throughout hypothermic storage. While discocytes are considered healthy RBCs, upon storage, they progressively convert into sphero-echinocytes and echinocytes due to loss of the membrane and reduced surface area (Fig. 4a). Storage-aged RBCs have fewer discocytes and more sphero-echinocytes. Therefore, we investigated whether intermittent DAMPs scavenging either on the 21st or 28th day by *Tau-Acr*NFS can prevent/reduce the transformation of discocytes into sphero-echinocytes. Fresh RBCs and storage-aged RBCs (42nd day) were imaged through a field-emission scanning electron microscope

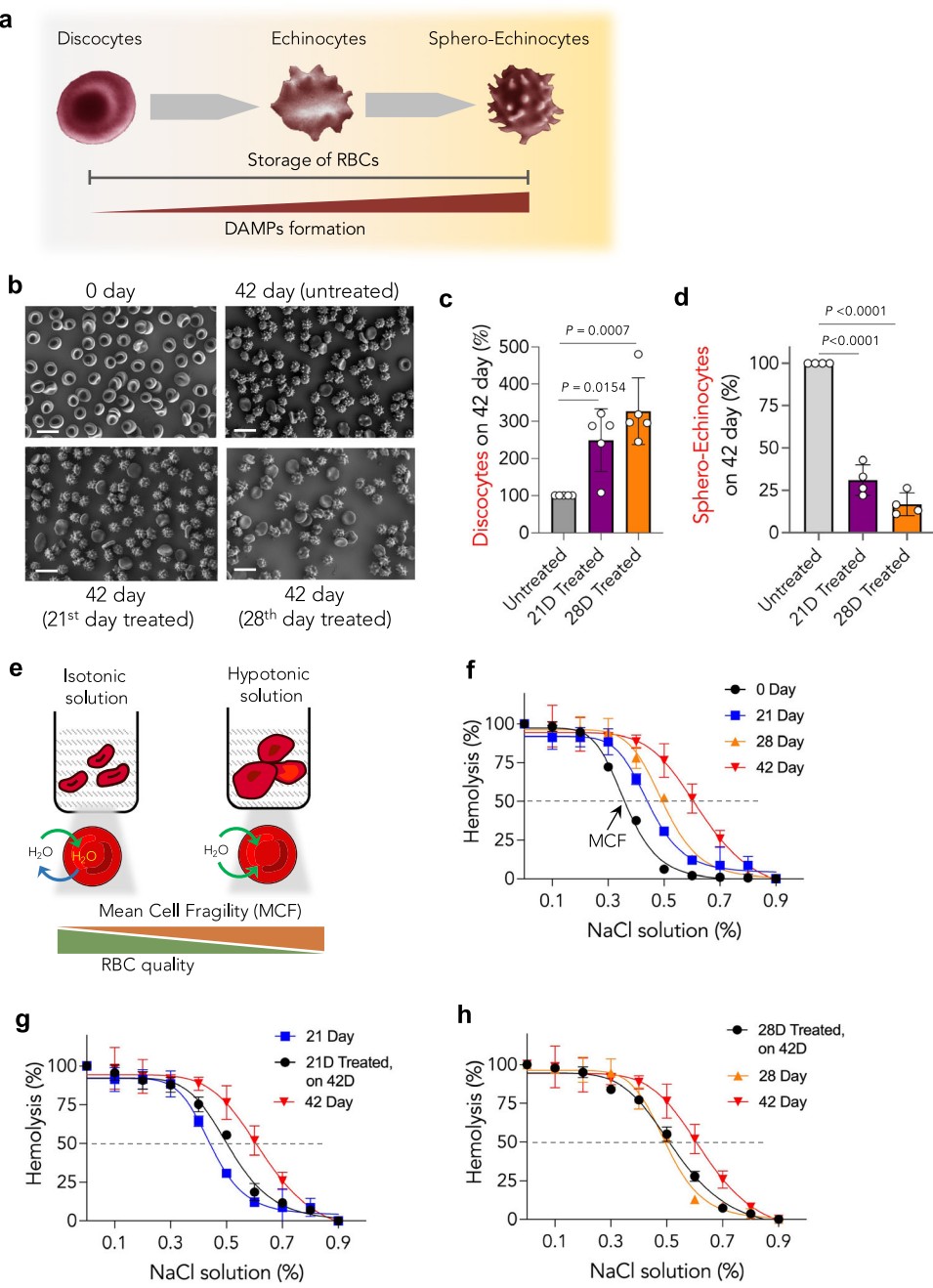

**Fig. 4 | Intermittent scavenging of DAMPs using *Tau-Acr*NFS prevents structural and morphological damage to stored human RBCs. a** Schematic of storage-induced structural transformation of healthy discocytes into non-healthy echinocytes and sphero-echinocytes. **b** SEM images of freshly collected RBCs (0th day), and storage-aged RBCs on the 42nd day. RBCs were stored for 42 days either without treatment or intermittently treated with *Tau-Acr*NFS either on the 21st or 28th day (**b**). Scale bar = 10 μm. The relative number of discocytes present on the 42nd day (**c**) revealed that intermittent scavenging of DAMPs using *Tau-Acr*NFS has increased by 250% and 350% for the 21st and 28th day treated RBCs, respectively (*n* = 5, from independent experiments). The relative number of sphero-echinocytes present on the 42nd day (**d**) revealed that intermittent scavenging of DAMPs using *Tau-Acr*NFS has decreased their number to 30% and 15% for the 21st and 28th day treated RBCs, respectively (*n* = 4, from independent experiments). The number was calculated from 10,000 RBCs in each sample. **e**–**h** Osmatic fragility test revealed that intermittent treatment had enhanced membrane integrity of stored RBCs. Schematic of osmatic fragility test to measure the mean cell fragility (MCF) by placing RBCs in a hypotonic solution, where RBC swells and undergoes hemolysis (**e**). The lower MCF values are equivalent to higher membrane integrity. MCF values of stored RBCs progressively increased as a function of storage time (**f**), suggesting that loss of membrane integrity of RBCs during the storage. MCF values of 42 days of stored RBCs that were intermittently treated were lower than untreated stored old RBCs (**f**, **g**), suggesting that intermittent removal of DAMPs reduced the damage of RBCs and enhanced the membrane integrity of stored old RBCs (*n* = 3, from independent experiments). Data in **c**, **d**, **f**–**h** are mean ± s.d., *P* values in data **c** and **d** were determined by ordinary one-way ANOVA with Tukey's post hoc analysis using GraphPad PRISM 9, and exact *P* values are indicated. Source data are provided as a Source Data file.

and estimated the number of discocytes and sphero-echinocytes. The images revealed that most RBCs were transformed into either sphero-echinocytes or spherocytes, while healthy discocytes reduced significantly (Fig. 4b).

The number of discocytes present in untreated 42 days of stored RBCs sample was considered 100% (counted from 10,000 cells, Fig. 4c). Similarly, the number of discocytes and sphero-echinocytes were counted in 42 days of stored RBCs samples that were untreated

and intermittently treated with *Tau-Acr*NFS on either the 21st or 28th day (Fig. 4b–d). The intermittent treatment has increased total discocytes on the 42nd day by 250 and 350%, respectively (Fig. 4c). On the contrary, compared to untreated samples, intermittent treatment on the 21st or 28th day has decreased the total number of spheroechinocytes on the 42nd day to 30 and 15%, respectively (Fig. 4d). These results suggest that intermittent removal of DAMPs using *Tau-Acr*NFS reduces the conversion of healthy discocytes into spheroechinocytes, hence, improves the quality of stored RBCs.

Furthermore, we have conducted the osmotic fragility test to characterize RBC membrane integrity by subjecting stored RBCs to a hypotonic solution (Fig. 4e). Mean cell fragility (MCF) is the concentration of NaCl at which 50% hemolysis occurs. The quality of the RBC membrane is inversely proportionate to the MCF value. We have observed 0 and 100% hemolysis in 0.9% (isotonic) and 0.1% (hypotonic) NaCl solutions, respectively. The MCF values for 0, 21, 28, and 42 days of stored RBCs were 0.36, 0.45, 0.5, and 0.64% NaCl, respectively (Fig. 4f), suggesting that the membrane integrity of RBCs progressively deteriorates upon storage. Interestingly, MCF values for 42-day RBCs, which were intermittently treated with *Tau-Acr*NFS either on the 21st or 28th day, were 0.5 and 0.51%, respectively (Fig. 4g, h), suggesting that intermittent removal of DAMPs slows down the deterioration of RBC membrane quality.

### Transfusion quality of *Tau-Acr*NFS treated stored old RBCs is comparable to freshly collected RBCs

Post RBCs transfusion, measuring the duration of transfused RBCs remaining in circulation (in vivo recovery) is a gold standard for quantifying the quality of stored RBCs[29]. A higher % of RBCs stay longer in circulation, which signifies the better quality of transfused RBCs. According to the FDA guidelines, transfusion of stored human RBCs is considered as successful when ≥75% of transfused human RBCs survive in circulation after 24 h[29]. Hence, similar standards were adopted to quantify the efficiency of the transfusion of stored mouse RBCs into mice[30]. Therefore, as a proof-of-concept, we measured the transfusion efficiency of *Tau-Acr*NFS treated stored mouse RBCs into mice, in vivo.

The shelf life of C57BL/6J mice RBCs is reported to be 14 days, equivalent to 42 days stored human RBCs[31]. First, we demonstrated that stored mouse RBCs generate DAMPs, such as cell-free Hb and cell-free nucleosomes, as a function of storage time (Fig. 5a, b). Subsequently, intermittent scavenging of DAMPs from stored mouse RBCs on the 5th and 10th days using *Tau-Acr*NFS reduced the production of DAMPs on the maximum allowed storage time, i.e., 14 days (Fig. 5c, d). To optimize intermittent treatment for RBC recovery experiments, mouse RBCs were intermittently treated with *Tau-Acr*NFS either once (on the 5th day or 10th day) or twice (on the 5th and 10th days), and DAMPs were measured on the 14th day (Supplementary Fig. 4). Data in Supplementary Fig. 4 suggest that total DAMPs on 14th day were significantly lower in samples that were treated twice (5th and 10th days), compared to samples treated once on either 5th or 10th day. Therefore, for RBC recovery experiments, stored mouse RBCs were intermittently treated on the 5th and 10th days. Cumulatively, these results suggest that intermittent scavenging of DAMPs might prevent progressive damage to RBCs and enhance transfusion efficiency.

In RBCs transfusion experiments, we labeled RBCs with biotin, in vivo. Intravenous administration of sulfo–NHS–biotin covalently binds and labels RBC membrane proteins with biotin[32]. After 48 h of post-administration of sulpho-NHS-biotin (1 mg/mouse), blood was collected from the same mouse. The biotinylated RBCs were transfused into a different mouse to study transfusion efficiency (Fig. 5e). Blood was collected at 1, 24, 48, and 72 h post-transfusion of biotin-labeled RBCs. The collected RBCs were complexed with Streptavidin APC-eFluor™ and quantified by flow cytometry to measure the % of labeled transfused RBCs that remain in circulation. Interestingly, ~95% of transfused fresh RBCs have remained in circulation for 72 h (Fig. 5f,

Supplementary Fig. 5), suggesting that freshly collected RBCs have the highest quality. To test the effect of storage, the biotin-labeled mouse RBCs were stored for 14 days, either without treating or intermittently treating with *Tau-Acr*NFS on the 5th and 10th days (Fig. 5e). After 14 days, biotinylated RBCs were transfused into mice, and tracked their circulation time. A total of 90% of untreated transfused RBCs were in circulation for 24 h, which was reduced to <40% at 48 h (Fig. 5g, Supplementary Fig. 6), suggesting that the quality of stored old RBCs deteriorated significantly. On the contrary, *Tau-Acr*NFS treated stored old RBCs remain >90% in circulation even at 72 h, similar to fresh RBCs (Fig. 5h, Supplementary Fig. 7), suggesting that intermittent treatment slows down stored RBC aging, and prevents their clearance from circulation by the spleen. Remarkably, these findings indicate that intermittent scavenging of DAMPs reduced the deterioration of stored RBCs, and the quality of 14 days-stored old RBCs is equivalent to freshly collected RBCs.

### Intermittent scavenging of storage lesion with *Tau-Acr*NFS enhances the shelf-life of stored old RBCs

In situ-generated DAMPs deteriorate RBC quality during storage, limiting their maximum shelf-life to 42 and 14 days for human and mouse RBCs, respectively. Intermittent removal of DAMPs by *Tau-Acr*NFS treatment enhanced the quality of RBCs; therefore, we envisaged that enhanced quality of RBCs may increase the shelf-life of stored old RBCs.

Initially, the phospholipid architecture of RBCs was studied to understand their shelf-life. Storage-associated loss of phospholipid asymmetry results in phosphatidylserine (PS) exposure in the outer leaflet of the RBC membrane[33–35]. PS exposure serves as an 'eat-me' signal to phagocytes to clear the RBCs from circulation[33,36] (Fig. 6a). PS exposure increases with storage and is linked with DAMPs production[37], hence, the low survivability of stored RBCs in circulation. Therefore, the number of RBCs expressing PS on the outer leaflet of their membrane was determined as described elsewhere[38]. In untreated stored RBCs, we found that 4, 20, and 40% of PS-exposed RBCs were present in 5, 14, and 17 days of stored mouse RBCs, respectively (Fig. 6b and Supplementary Fig. 8). On the contrary, intermittent scavenging of DAMPs by *Tau-Acr*NFS has significantly reduced PS-exposed RBCs on days 14 and 17 (Fig. 6c and Supplementary Fig. 9). Hence, the overall quality of RBCs is higher in the intermittently treated samples even on the 17th day, which is three days longer than the maximum allowed 14 days.

Furthermore, we have conducted RBC recovery experiments using long-term stored mouse RBCs. As described in the previous section, biotinylated murine RBCs were stored for up to 18 days either without treating or intermittently treated with *Tau-Acr*NFS on the 5th and 10th days. The different days of stored biotinylated RBCs (15–18 days) were transfused into mice, and measured the percentage of transfused RBCs in circulation (Fig. 6d–g and Supplementary Figs. 10–16). As expected, when DAMPs were not scavenged, 15–17 days-stored RBCs could not remain longer in circulation, and the number of transfused RBCs reduced to <30% within 24 h, which is well below the accepted 75% range. On the contrary, 15–17 days-stored RBCs when they were intermittently treated with *Tau-Acr*NFS, post-transfusion, >95% of transfused RBCs remained in circulation after 24 h (Fig. 6d–f). However, whether treated or not, 18 days-stored RBCs could not remain in circulation for 24 h (Fig. 6g), suggesting that the upper limit for storing the intermittently treated RBCs is 17 days. Therefore, the maximum shelf life of murine RBCs was increased from 14 days to 17 days.

Post-blood transfusions, DAMPs are known to cause systemic immunomodulation. Hence, we tested the efficacy of *Tau-Acr*NFS treatment to prevent transfusion-induced immunomodulation. To test that, we have transfused fresh RBCs (0 day), untreated and intermittently treated 17 days-stored RBCs in C57BL/6J mice (8–12 weeks).

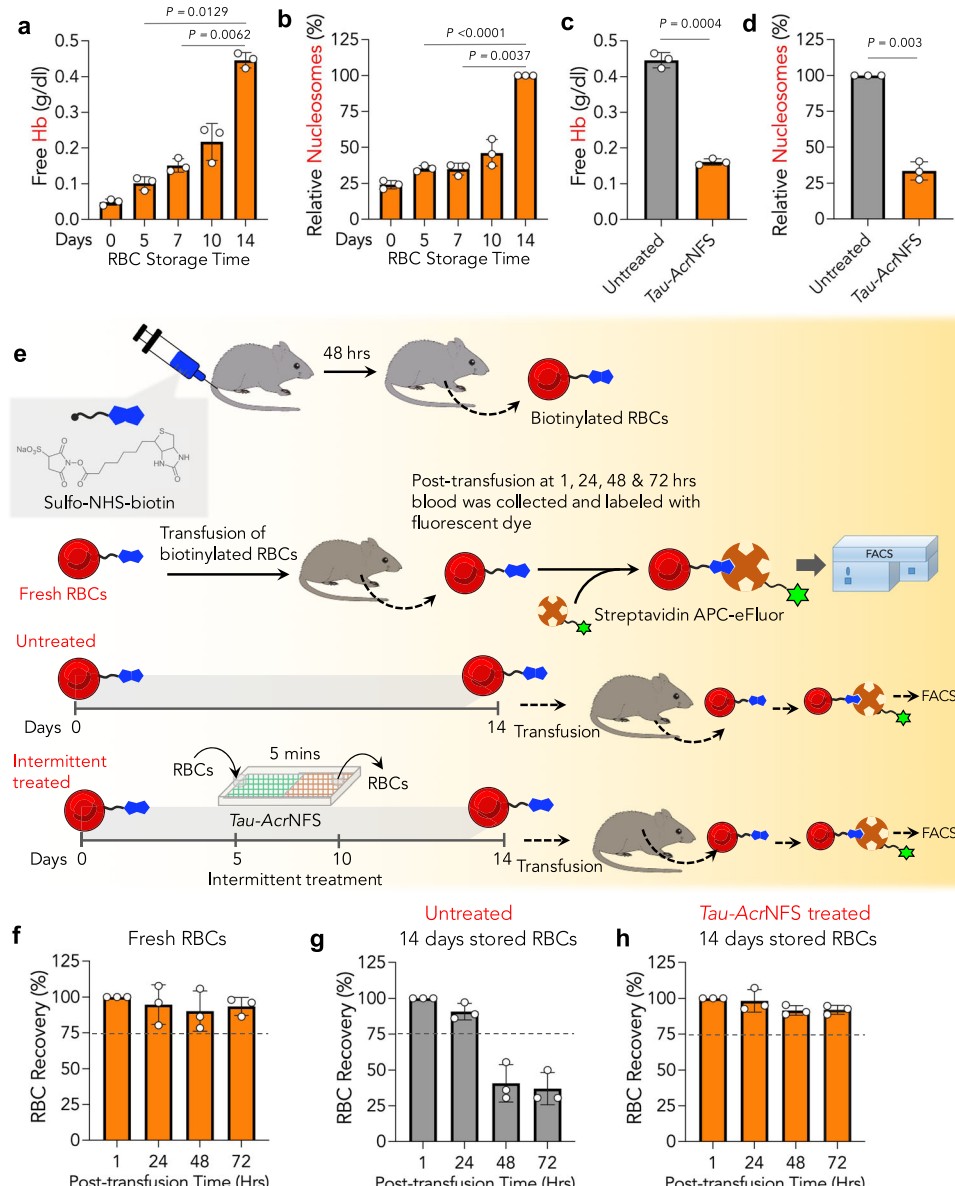

**Fig. 5 | Mice RBCs generate DAMPs during their storage, and intermittent treatment with *Tau-Acr*NFS enhanced the quality of stored old RBCs equivalent to fresh RBCs in vivo. a, b** Stored mice RBCs produced DAMPs such as free hemoglobin and nucleosomes. **c, d** The relative concentration of DAMPs before and after *Tau-Acr*NFS treatment has significantly reduced accumulated Hb (**c**), and nucleosomes (**d**). **e–h** Biotinylation of RBCs, and post-transfusion recovery of RBCs, in vivo. Schematic of RBC recovery experiments, in vivo (**e**). After 48 h of intravenous administration of Sulfo–NHS–biotin (1 mg/mouse) in C57BL/6J mice (8–12 weeks), biotin-labeled RBCs containing blood were collected. In the fresh RBCs recovery group, freshly collected biotinylated RBCs were transfused into C57BL/6J mice (8–12 weeks) and collected the blood post-transfusion times 1, 24, 48, and 72 h. The collected RBCs were labeled with Streptavidin APC-eFluor™ and quantified via flow cytometry to measure the % of labeled transfused RBCs that remain in circulation. The number of labeled RBCs present in circulation has been

quantified (**f**). Similarly, 14 days-stored biotinylated RBCs, either untreated or intermittently treated with *Tau-Acr*NFS, were transfused into mice, collected blood at post-transfusion 1, 24, 48, and 72 h, and labeled with Streptavidin APC-eFluor™, and quantified via flow cytometry (**g** and **h**, $n = 3$, from independent experiments). In untreated stored RBCs, ~90% of RBCs were in circulation for 24 h but rapidly cleared thereafter (**g**). On the contrary, post-transfusion of intermittently treated RBCs, >90% of transfused-RBCs were found in circulation even at 72 hrs (**h**), suggesting that intermittent treatment has increased the quality of stored old RBCs equivalent to fresh RBCs. Data in **a**–**d**, and **f**–**h** are mean ± s.d. ($n = 3$, from independent experiments). For **a**, **b**, *P* values were determined by Repeated Measures one-way ANOVA, and for (**c**, **d**), by Student's *t*-test with Welch's correction using GraphPad PRISM 9, and exact *P* values are indicated. Source data are provided as a Source Data file.

Two hours post-transfusion, blood was collected, and cytokines/chemokines in plasma were quantified using Mouse Cytokine Array Panel A (Methods, Fig. 6h, Supplementary Fig. 17). Transfusion of untreated 17 days-stored RBCs has elevated expression of stromal cell-derived factor1 (CXCL12), macrophage-colony-stimulating factor (M-CSF), complement component 5A(C5A), and keratinocyte-derived chemokine/CXCL1 (KC/CXCL1). Transfusion of untreated 17 days-stored RBCs has led to 2–3 fold higher cytokine production levels than either fresh

RBCs or intermittently treated 17-days-stored RBCs (Fig. 6h). Additionally, mice spleens were collected 2 h post-transfusion, and spleens were significantly darkened when transfused with untreated RBCs compared to treated RBCs (Fig. 6i). The darkening of the spleen is associated with iron deposition after transfusing poor quality RBCs, loaded with free Hb, which corroborated by $Fe^{2+}$ quantification using ICP spectroscopy (Fig. 6j). These results suggest that intermittent scavenging of DAMPs using *Tau-Acr*NFS enhanced the quality of stored

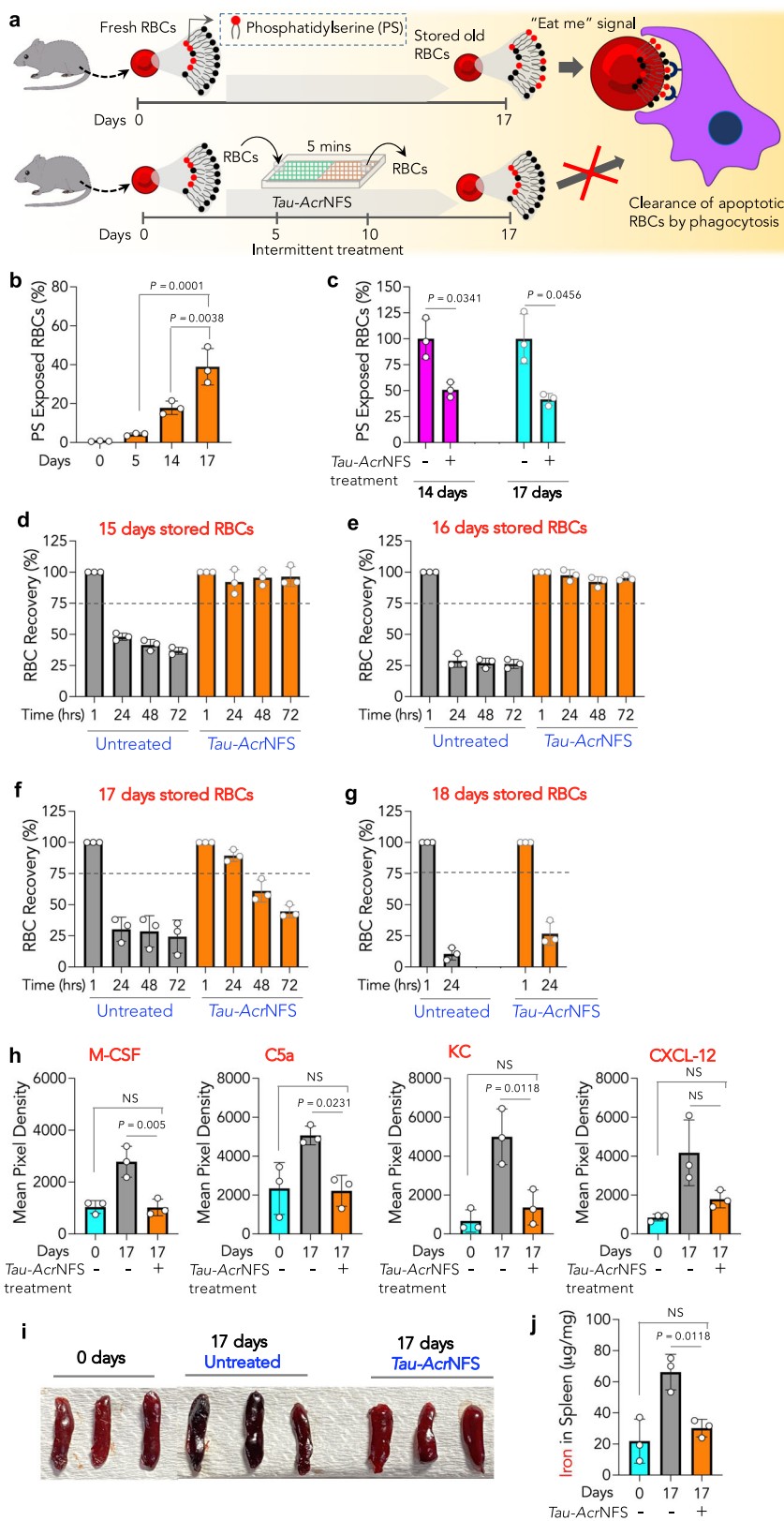

old RBCs equivalent to fresh RBCs and significantly enhanced their shelf-life.

## Discussion

Transfusion of RBCs is a lifesaving practice, which is limited by the progressive deterioration of stored RBCs as a function of time. In situ

generated DAMPs such as cell-free DNA, cell-free nucleosomes, free Hb, free iron, and polyunsaturated fatty acids are the primary cause of declining RBCs condition. Large observational studies have reported a mean duration of RBC storage of between 16 and 21 days, with a maximum storage duration before transfusion is limited to 42 days with standard preservative solutions[39]. Until now, existing strategies are

**Fig. 6 | Intermittent scavenging of DAMPs using *Tau-Acr*NFS enhances the shelf-life of stored old RBCs and prevents systemic inflammation in vivo.**
**a**–**c** During the storage, exposure of phosphatidylserine (PS) on the outer leaflet of the RBC membrane serves as an 'eat-me' signal, thus, RBCs will be eliminated from circulation by phagocytes. Intermittent scavenging of DAMPs by *Tau-Acr*NFS reduces PS exposure, therefore, keeping RBCs in circulation for longer periods (**a**). Time-dependent PS exposure on mice RBCs (**b**), intermittent treatment reduced PS exposure on RBCs (**c**). **d**–**g** Recovery of storage-aged RBCs either untreated or intermittently treated with *Tau-Acr*NFS on the 5th & 10th days. The maximum allowed storage time for mice blood is 14 days. Recovery data suggests that the quality of untreated RBCs that are stored for 15–17 days declines drastically, and RBCs are eliminated from circulation immediately after transfusion. Only 15–20% of transfused RBCs were found in circulation after 24 h (**d**–**f**), which is well below the accepted 75%. On the contrary, intermittently treated RBCs, even after storing for

17 days, >95% of transfused RBCs were found in circulation at 24 hrs (**d**, **e**, and **f**), suggesting that intermittent removal of DAMPs has decreased RBC damage, and enhanced the shelf-life of RBCs at least by 22%. However, irrespective of whether intermittent treatment or not, 18 days-stored RBCs were eliminated from circulation, with only 25% of RBCs found after 24 h (**g**), suggesting the upper limit of intermittent treatment. **h**–**j**, Fresh RBCs, or 17 days-stored RBCs, either untreated or intermittently treated, were transfused into C57BL/6J mice (8–12 weeks). Two hours of post-transfusion, serum inflammatory cytokines were quantified (**h**), spleens were collected and imaged (**i**), and iron in the spleens was quantified using ICP-AES (**j**). Data are mean ± s.d. (*n* = 3, from independent experiments). For **b**, **h**, **j**, *P* values were determined by ordinary one-way ANOVA with Tukey's post hoc analysis and for (**c**), by two-tailed Student's *t*-test with Welch's correction using GraphPad PRISM 9, and exact *P* values are indicated. NS = not significant. Source data are provided as a Source Data file.

limited to developing preservatives and rejuvenating solutions. Additionally, engineered-proteins-containing filters have been developed to absorb pathogen-specific toxins[40,41]. However, there was no effort made to scavenge DAMPs during RBC storage to prevent RBC damage. Interestingly, these DAMPs carry either cationic or anionic charges[42]. Therefore, we have designed polymers with an appropriate hydrophilicity/hydrophobicity balance, and carrying anionic (due to taurine) or cationic (due to acridine) charges. Robust electrospun nanofibrous sheets (*Tau-Acr*NFS) were generated using these polymers. The charged *Tau-Acr*NFS do not cause hemolysis, and upon incubation, RBCs or any other cells do not adhere to these sheets, suggesting that *Tau-Acr*NFS are robust and consist of key parameters that play a critical role in translating this technology into a medical device.

Interestingly, scavenging DAMPs using *Tau-Acr*NFS has a dual advantage. When *Tau-Acr*NFS treatment is used terminally at the end of the storage, they efficiently scavenge in situ generated DAMPs from stored blood. On the other hand, intermittent treatment of stored blood using *Tau-Acr*NFS can prevent the damage of RBCs, hence, significantly lowering the production of DAMPs until their maximum storage time. We have identified a suitable storage day for intermittent treatment through systematic time-dependent experiments. We found that 28th-day treatment results in a maximum reduction of DAMPs and retains healthier RBCs on the 42nd day. This observation could be because the accelerated deterioration of RBCs majorly occurs between the 28th day and to 42nd day of storage.

The structural integrity of RBCs plays a vital role in increasing their post-transfusion circulation time. The osmotic fragility measurements revealed that compared to untreated RBCs, removal of DAMPs on the 28th day using *Tau-Acr*NFS enhanced 42 days stored human RBC's membrane integrity and reduced PS-exposure on the outer leaflet of the membrane to reduce 'eat-me' signals. It was envisaged that maintaining membrane integrity should lead to extended survival in circulation. Post-transfusion recovery of murine RBCs affirmed this hypothesis. Murine RBCs can be stored a maximum of 14 days, akin to 42 days for human RBCs. Remarkably, mouse RBCs recovery suggested that intermittent removal of DAMPs enhanced the quality of 14 days-stored old RBCs equivalents to freshly collected RBCs. In addition to improving the quality, intermittent removal of DAMPs has increased the shelf-life by at least 22%. Systemic immunomodulation is one of the hallmarks of the transfusion of stored old RBCs. As we demonstrated here, intermittent removal of DAMPs has lessened the accumulation of DAMPs in stored old RBCs and ultimately reduced immunomodulation. Therefore, it is anticipated that this approach would reduce transfusion-related complications. We are designing a *Tau-AcrNFS*-based insert in blood bags as a medical device, which will be feasible to use in the blood bank processing setting without the need for multiple interventions during the storage of RBCs.

In summary, our study implicates in situ generated DAMPs in causing accelerated deterioration of RBCs. We establish a novel

approach to slow down the damage of stored RBCs through intermittent scavenging of the storage lesion. In the future, treating with the nanofibrous sheets before the transfusion could significantly reduce DAMPs and prevent transfusion-related complexities. Furthermore, the intermittent treatment adds a new dimension to maintaining RBC quality. Finally, our investigation establishes the concept of scavenging DAMPs during storage, leading to increased quality and shelf-life of RBCs. Furthermore, the use of charged electrospun nanofibrous sheets may lead to the development of novel blood bags or an insert-based medical device. Therefore, it may significantly impact healthcare by improving the RBC transfusion quality.

## Methods
### Ethics
Our research involved in preclinical study and human volunteers for blood collection complies with all relevant ethical guidelines. Human blood was collected according to the approved protocols from the Institutional Human Ethical Committee (inStem/IEC-10/003) of the Institute for Stem Cell Science and Regenerative Medicine (inStem). Informed consent has been obtained from the participants. Blood donors are both male (*N* = 3, age 26–30 years) and female (*N* = 3, age 25–28 years). All mouse studies strictly adhered to institutional and national guidelines for humane animal use. The experimental protocols were approved by the Institutional Animal Ethics Committee (IAEC) at the Institute for Stem Cell Science and Regenerative Medicine (INS-IAE-2020/16(R1)).

### Materials
Poly (methoxy vinyl ether-alt-maleic anhydride) average Mw-330,000 (PMVEMA), was used in all synthesis procedures. Tetrahydrofuran (THF), tetradecylamine, taurine, sodium bicarbonate, ethyl acetate, 3,6-bis(dimethylamino) acridine hemi (zinc chloride) salt, benzene, methanol, and ammonium hydroxide, *N,N*-dimethyl ethylene diamine, 1,6-dibromohexane, sodium sulfate, toluene, dimethylformamide (DMF), triton X-100, chamber slides(HI Media), blood collection tubes (BD Vacutainer), citric acid monohydrate, trisodium citrate dihydrate, sodium dihydrogen phosphate, dextrose, sodium chloride, glucose anhydrous, mannitol, adenine, APTMS (3-(Aminopropyl)trimethoxysilane), glutaraldehyde, cacodylate buffer, osmium tetroxide, hexamethyldisilazane, ethanol, double distilled water and deionized water were used. Unless mentioned otherwise, all chemicals were procured from Sigma-Aldrich.

### Synthesis of anionic and cationic scaffolds
A detailed scheme for the synthesis is described in Supplementary materials.

**PMVEMA -C₁₄**. The PMVEMA (500 mg, 3.2 mmol) was added to tetrahydrofuran (THF) (10 ml) in a pressure tube and stirred at 80 °C until dissolution. Subsequently, tetradecylamine (345 mg, 1.6 mmol) in THF

(10 ml) was added to the previous mixture. The reaction mixture was heated at 80 °C for 3 h. The solvent was finally removed using a rotavapor to get the polymer in dry powder form (800 mg, ~90% yield). *$^1$H-NMR (CD$_3$OD, 600 MHz)*, δ: 3.0-3.51 (Br, 9H, -OCH$_3$, -O-CH, PMVEMA and -NH-CH$_2$ alkyl protons), 1.47 (Br, 2H, alkyl protons), 1.22 (Br, 22H, alkyl protons), 0.845 (t, 3H, alkyl protons). As per NMR data, ~50% of PMVEMA polymer was substituted with tetradecyl amine. *FTIR (cm$^{-1}$):* 3354, 2926, 1858, 1782, 1733, 1697

**Poly-Taurine: PMVEMA-C$_{14}$-Tau.** A solution of taurine (100 mg, 0.8 mmol) and sodium bicarbonate was prepared (84 mg, 1.0 mmol) in deionized water (1 ml). This solution was added dropwise to the solution of PMVEMA-C$_{14}$(50%) (800 mg) in THF (20 ml) and left for stirring at 80 °C for 16 h. The reaction mixture was subjected to rotavapour to concentrate. Subsequently, it was sequentially triturated with ethylacetate and deionized water to remove the residual tetradecylamine & taurine. The final polymer was obtained in a gel form, which was lyophilized to a white powder (720 mg, ~75% yield). *$^1$H-NMR (CD$_3$OD, 800 MHz)*, δ: 3.9-4.1 (Br, 1H, taurine), 2.7-3.5 (Br, 5H, -OCH$_3$, -O-CH, PMVEMA and -NH-CH$_2$ alkyl protons), 1.95 (Br, 2H, PMVEMA protons), 1.4 (Br, 1H, alkyl protons), 1.1-1.2 (Br, 11.5H, alkyl protons), 0.8 (t, 1.5H, alkyl protons). As per the NMR data, PMVEMA was approximately substituted by 50% of C$_{14}$H$_{29}$-NH$_2$ and ~25% of taurine. *FTIR (cm$^{-1}$):* 3209, 3046, 2946, 1722, 1616, 1037.

**Acridine hexyl bromide.** 3,6-bis(dimethylamino) acridine hemi (zinc chloride) salt (10 g, 27.02 mmol) was dissolved in methanol (50 ml), and excess ammonium hydroxide (50 ml) was added. Acridine orange (AO) base was extracted from benzene and dried over sodium sulfate; the AO base (5 g) was recovered from benzene using rotavapor. The AO base (2.65 g, 10 mmol) was dissolved in anhydrous toluene (100 ml), and 1,6-dibromohexane (7.7 ml, 50 mmol) was added to the solution and refluxed at 120 °C overnight. The bright red precipitate in the reaction mixture was filtered and dried (yield- 2.1 g, 41.2%). *$^1$H-NMR (CDCl$_3$, 600 MHz)* δ: 8.7 (s, 1H), 7.9 (d, 2H), 7.08 (m, 2H), 6.75 (s, 2H), 5.0 (t, 2H), 3.4 (t, 2H), 3.3 (s, 12H), 2.048 (m, 2H), 2.029 (m, 2H), 1.96 (m, 2H), 1.94 (m, 2H).

**Poly-Acridine**
**PMVEMA-C$_{14}$-Acr.** The solution of *N,N*-dimethyl ethylene diamine (87 μl, 0.8 mmol) in THF (10 ml) was added dropwise to the solution of PMVEMA-C$_{14}$(50%) (800 mg) in THF (20 ml) in a pressure tube. The mixture was allowed to stir at 80 °C for 12 h. After its dissolution, THF in the reaction mixture was reduced until a viscous solution appeared using rotavapour. Dry DMF (10 ml) was added to this viscous solution, followed by the addition of Acridine-hexyl bromide (243 mg, 0.48 mmol) in DMF (5 ml). The resulting mixture was stirred at 50 °C for 16 h. The polymer was triturated thoroughly with deionized water to remove unreacted Acr-Hexyl-Bromide and DMF. Further, the polymer was lyophilized to an orange powder (800 mg, ~80%) *$^1$H-NMR (CDCl$_3$, 800 MHz)*, δ: 6.5-8.6 (Br, 0.77H, acridine protons), 3.0-3.5 (Br, 4H, PMVEMA protons), 0.98-1.7 (Br, 9.5H, alkyl protons), 0.87 (Br, 3H, alkyl protons). As per the NMR data, approximately 50% of tetradecylamine (C$_{14}$H$_{29}$-NH$_2$) and ~11% of acridine hexyl bromide were substituted on PMVEMA polymer.

## Fabrication of electrospun nanofibrous sheets
The electrospinning solutions were prepared by dissolving *poly*-Taurine (20%, w/v) and *poly*-Acridine (15%, w/v) in dry DMF at room temperature (RT). These solutions were stirred overnight before electrospinning. During the electrospun procedure using the Espin device (ES2, Espin nanotech, India), a 2-mL plastic syringe (Dispo Van, India), and a blunt 22 G needle (Dispo Van, India) were used. The solutions were pumped via a syringe pump at a constant flow rate of 0.5 ml/h (*poly*-Taurine) and 0.3 ml/h (*poly*-Acridine). The electrospun

nanofibrous sheets (*Tau*-NFS and *Acr*-NFS) were collected on an aluminum foil placed 15 cm away from the needle (where 15 kV voltage was applied). All experiments were carried out at 25 °C and less than 55% relative humidity (RH). The electrospun mats with a length and width of approximately 10 cm × 10 cm were stored in a desiccator until further use.

## Field emission scanning electron microscopy
The morphology of the *Tau*-NFS and *Acr*-NFS was investigated by FESEM (Carl Zeiss MERLIN VP compact). Dried electrospun mats were mounted on aluminum stubs and sputter-coated with gold (Pelco® SC-7 Auto sputter coater). The gold-coated mats were imaged at a voltage of 2 kV. The scaffolds treated with pRBCs were imaged to investigate the adherence of blood cells.

## Contact angle measurement
The wettability of *Tau*-NFS and *Acr*-NFS was studied using the sessile drop method. The scaffolds were cut uniformly and placed on a glass slide before the measurement. Water droplets (5 μL) ~2 mm in diameter were dropped on the electrospun NFS, and the images were acquired. The images were processed using ImageJ software (V1.53).

## Hemolysis assay
A hemolysis assay was performed to evaluate the hemolytic nature of *Tau-Acr*NFS. The blood was collected in a tube containing anticoagulant sodium citrate. The collected blood was centrifuged (500 G, 10 min), and the supernatant was discarded. These washed RBCs were used to prepare 2% hematocrit in saline and added to *Tau-Acr*NFS fixed in a chamber slide (16 cm$^2$) prewashed with double distilled water. The RBCs were further resuspended in saline and pelleted down. This setup was kept on an incubator shaker (37 °C and 100 rpm) for 1 hour. After 1 hour, the hematocrit was transferred into a 2 ml centrifuge tube and centrifuged at 500 G for 10 min at RT. A 100% lysis sample of the untreated RBC specimen was prepared with 0.1% Triton-X100 in 2% hematocrit. The concentration of Hb in a 100% lysed RBCs sample was determined by cyanmethemoglobin (Drabkin's) method, which is considered as 100%. Afterward, a series of dilutions was prepared from lysed sample to generate a standard curve. The level of hemolysis of the *Tau-AcrNFS*-treated RBCs is then derived from the standard curve using Drabkin's method. For Drabkin's method, the absorbance was recorded at 540 nm using Varioskan LUX Multimode Microplate Reader (ThermoFisher, Massachusetts, USA).

## Polymer leaching test
*Acr*-NFS (8 cm$^2$) was fixed on the chamber slide and incubated in Optisol (AS-5) solution [composition: sodium chloride (8.77 mg/mL), glucose anhydrous (8.18 mg/mL), mannitol (0.01% w/v) and adenine (0.03 mg/mL)]. The sheets were incubated for 2 h. The optisol solution was removed after both incubations and lyophilized to recover the leached *poly*-Acridine from the sheets. The lyophilized *poly*-Acridine was dissolved in DMF and estimated using a fluoro spectrophotometer (Horiba Fluro log QM, France). Fluorescence intensities were recorded as a function of time with excitation at 502 nm and emission at 535 nm. The Leaching (%) was determined via a pure *poly*-Acridine polymer standard curve prepared at the excitation and emission wavelengths mentioned above.

## Packing of human RBC
Blood was collected according to the approved protocols from the Institutional Human Ethical Committee (inStem/IEC-10/003) of the Institute for Stem Cell Science and Regenerative Medicine (inStem). Informed consent has been obtained from the participants. RBC units were prepared from human whole blood (both male and female, *n* = 3) of different blood groups collected in 20 ml vacutainers containing citrate-phosphate-dextrose with adenine (CPD-A) that contains citric

acid monohydrate (3.577 mg/mL), trisodium citrate dihydrate (29.972 mg/mL), sodium dihydrogen phosphate (2.496 mg/mL) and dextrose (25.5 mg/mL). The whole blood was centrifuged for 20 min at 500$g$, and the supernatant was removed. The removal of supernatant was followed by the addition of optisol (AS-5, preservative) that contains sodium chloride (8.77 mg/mL), glucose anhydrous (8.18 mg/mL), mannitol (0.01% w/v) and adenine (0.03 mg/mL) to obtain a hematocrit of ~65%, which was transferred into standard PVC bags, and stored at 4 °C in the dark.

### Scavenging of DAMPs from stored human RBCs using *Tau*-NFS and *Acr*-NFS, in vitro

On different days of, stored RBCs (0, 14, 21, 28, and 42) were incubated with *Tau*-NFS and *Acr*-NFS individually and in combination for 5 min in total to scavenge DAMPs. In both treatment conditions, the nanofibrous sheets were prewetted with distilled water for 30 min before the addition of RBCs. The scavenging experiments were performed on ice under the aseptic condition. *Tau-Acr*NFS was mounted on the chamber slides, and the blood samples were added. It was made sure that the blood should cover the entire surface area and be in contact with nanofibers. The total surface area of the scaffold was kept at 24 cm$^2$ for each experiment. Scaffold treatment of RBCs was done at 4 °C for 5 min to keep the cold chain storage.

### Quantification of free Hb using Cyanmethemoglobin (Drabkin's) method.

Drabkin's reagent lyses RBCs and oxidizes all forms of Hb, except for the minimally present sulfhaemoglobin, to the stable HiCN[43]. The supernatant of the stored RBCs unit (scaffold-treated or untreated) was diluted (1:10) with Drabkin's reagent (Sigma, St. Louis, MO, USA). Human Hb diluted with Drabkin's reagent (1:10) (Sigma Aldrich, USA) was used to prepare the standards (0–40 mg/ml) and the calibration curve. The samples and standards were incubated in the dark at RT for 15 min. Absorbance was recorded at 550 nm using a Varioskan LUX Multimode Microplate Reader (ThermoFisher, Massachusetts, USA). The Hb concentration of each sample was calculated from the human Hb calibration curve.

### Quantification of free DNA by PicoGreen assay.

RBC supernatants from scaffold-treated and untreated groups were thawed on ice. Supernatants were diluted (1:50) with PicoGreen® reagent and Tris-EDTA (TE) buffer (Quant-iT™ PicoGreen™ dsDNA Assay Kit, Invitrogen, USA), followed by 5 min incubation in the dark. Free-DNA estimation was performed using a fluoro spectrophotometer (Horiba Fluro log QM, France)[44].

### Quantification of nucleosomes.

Stored RBC supernatant was diluted in PBS (1:5). The Anti-Histone-Biotin-Monoclonal antibody (cloneH11-4) and Anti-DNA-POD- Monoclonal antibody from mouse (clone MCA-33) was used for quantifying nucleosomes by ELISA (Catalog No. 11920685001, Cell Death Detection kit, Roche, Indianapolis, IN), according to manufacturer's instructions.

### Osmotic fragility test

The stored RBC units (treated and untreated group) were added to 9 tubes containing saline concentration series ranging from 0% (distilled water) (Tube 1) to 0.9% (Tube 9). The tubes were gently mixed and incubated at room temperature for 30 min. The incubation was followed by centrifugation at 500$g$ for 20 min, and supernatants were collected. The optical density of the supernatant was measured by Varioskan LUX Multimode Microplate Reader (ThermoFisher, Massachusetts, USA) at 540 nm. Hemolysis in each tube was expressed in percentage. The maximum absorbance value of haemolyzed RBCs was taken at 100% in the distilled water. The erythrocytes treated with normal saline were used as a negative control (0% hemolysis).

### Primary and secondary fixation of RBCs for SEM

Glass coverslips were sonicated in acetone and cleaned. They were further dipped in 2% APTMS (3-(Aminopropyl)trimethoxysilane), and diluted in acetone to impart a coating of APTMS. RBCs were primarily fixed with 2% glutaraldehyde (GTA), incubated for 30 min in the dark, and washed with PBS. The primary fixed RBCs were incubated on APTMS-coated coverslips for 1 h. After 1 h, the coverslips were rinsed twice with PBS and incubated with 0.1 M cacodylate buffer for 10 min. Secondary fixation was achieved via 1% (w/v) Osmium tetroxide (OsO4) in 10 min to 16 hours. The secondary fixed coverslips were then rinsed with MilliQ water (10 min), followed by a dehydration step that involves a gradient ethanol wash, ranging from 40 to 100%. The dehydration step is followed by HMDS (Hexamethyldisilazane) drying. Finally, dried samples were mounted on aluminum stubs, sputter-coated with gold, and imaged under FESEM (Carl Zeiss MERLIN VP compact).

### Bioactive lipid estimation by LCMS

**Chemicals and standards.** Eicosanoids; 5(S)-HETE, 12(S)-HETE, 15(S)-HETE, and AA were purchased from Cayman Chemical Co. (Ann Arbor, Michigan, USA). 5(S)-HETE-d8 (Cayman Chemical Co) was used as an internal standard (IS) for this study. Acetonitrile (Baker analyzed® LC-MS reagent), water (Baker analyzed® LC-MS reagent), formic acid, ethanol, ethyl acetate (EtOAc), and glacial acetic acid were used in sample preparations for LC−MS/MS measurements.

**Preparation of stock solutions.** 5-(S)-HETE, 12(S)-HETE, 15(S)-HETE, AA, and IS 5(S)-HETE-d8 were constituted in ethanol by the provider. 5, 12 & 15 (S)-HETE was further processed and serially diluted to obtain solutions ranging from 0.05 to 100 ng/ml. AA solutions ranged from 0.5 to 1000 ng/ml. The working concentration of the pure IS was 1000 ng/ml.

**Sample preparation.** Lipid extraction from stored RBCs supernatant was performed by gently mixing the samples with 0.5 mL of EtOAc containing 0.13% acetic acid (v/v). All samples were spiked with 5 µl of the IS. After 10 min of vigorous shaking (1500 rpm), samples were centrifuged (10,000 rpm, 10 min, 4 °C), and 450 µl of the organic layer was transferred to fresh tubes and evaporated in speed vac (Scan Vac, Labogene, Demark) at 4 °C. The dry residue was reconstituted in 25 µL of nitrogen-purged EtOH and then directly injected (2 µl) into the LC−MS/MS system.

**LC–MS/MS conditions.** HPLC analysis was done using a Shimadzu Prominence HPLC system. The instrument was equipped with a communication module (CBM-20A), degassing unit (DGU-20A5R), solvent delivery units (LC-30AD), column oven (CTO-20AC), and an autosampler (SIL-30AC). The temperature of the column oven was maintained at 40 °C while the autosampler was set at 4 °C. The lipids were separated using an Acquity UPLC® BEH C18 1.7 µm column (2.1 × 50 mm). The mobile phase comprised solvent A, i.e., water with 0.1% formic acid (FA) and acetonitrile with 0.1% FA as solvent B. The flow rate was maintained at 0.2 ml/min, and the injection volume was 2 µl. The gradient for lipid elution (20.10 min) was set as 50% A and 50% B for one minute. This was followed by a linear gradient to 20% A and 80% B for 8 min to elute 5, 12, and 15 HETE. This was further followed by a linear gradient to 100% B for 2 min and kept for 5 min to elute AA. Finally, the column was equilibrated at 50% A and 50% B for 4 min before the next injection. LC-MS analysis was performed by QTRAP 5500 (Sciex, Framingham, Massachusetts, USA), a triple quadrupole mass spectrometer. LC−MS analysis was performed by MultiQuantTM 3.0.3 Software (SCIEX). The operating parameters for the mass spectrometer were as follows: declustering potential (DP): −209.8 V, entrance potential (EP): −5.7 V, collision cell exit potential (CXP): −16.9 V, ion spray voltage: −4500 V, source temperature: 500 °C,

curtain gas: 35 psi, ion source gas 1: 40 psi and ion source gas 2: 50 psi. All the spectra of lipids were recorded and quantified in multiple reaction monitoring (MRM) mode. Negative electrospray ionization was employed for all the lipids, and the deuterated IS. The ion Q1/Q3 transitions and other parameters can be seen in Supplementary Table 1.

### In vivo experimental design
**Mice.** Both male and female C57BL/6J mice of age 8–12-weeks-old were used for the experiments. The animals were housed in the animal facility at the National Center for Biological Sciences, Bengaluru. Animals were caged (maximum, four per cage) during the experiment, food and water were provided *ad libitum*. Mice were housed in facility that was subjected to 12/12 h light/dark cycle, at a temperature of 20–25 °C and humidity of 35–65%. All mouse studies strictly adhered to institutional and national guidelines for humane animal use. The experimental protocols were approved by the Institutional Animal Ethics Committee (IAEC) at the Institute for Stem Cell Science and Regenerative Medicine (INS-IAE-2020/16(R1)).

**Packaging and storage of mice RBCs and their treatment with Tau-AcrNFS to scavenge DAMPs.** Blood collected in CPD was immediately transferred and centrifuged at 500*g* for 20 min at 4 °C. Plasma was carefully aspirated, and the pRBCs were pooled together. Optisol (AS-5) was added to the pRBCs at 65% hematocrit. Totally 550 μL and 250 μL blood units were finally aliquoted from these pRBCs and stored in sterile Eppendorf tubes (0.6 and 0.3 ml). Mouse pRBCs were treated with *Tau-Acr*NFS (24 cm$^2$) for 5 min on the 5th and 10th day of storage at 40 °C. After the scavenging procedure, the RBCs were again transferred into autoclaved 2 ml Eppendorf tubes, leaving a small residual space, and stored till the 17th day. Its supernatant was taken for extracellular Hb and nucleosome estimation.

**Quantification of PS on RBCs by Annexin V (AV).** The number of erythrocytes expressing PS on their outer membrane was determined as described elsewhere[38]. Briefly, labeling with AV was performed by adding 5 μL of AV-Alexa Fluor™ 568 conjugate (Invitrogen, USA) in 10$^6$ erythrocytes in 100 μL Annexin binding buffer. The binding buffer was prepared by adding 140 mM and 2.5 mM CaCl$_2$ in 10 mM HEPES Buffer (all from Sigma-Aldrich). After incubation on ice for 15 min, cells were diluted four times with binding buffer and analyzed on a BD LSR Fortessa cytometer (BD Biosciences). Data was collected with FACSDiva v8.02, and data analysis was performed FlowJo v10.0.8. The erythrocyte population with exposed PS labeled with AV-Alexa Fluor™ 568 conjugate was measured as a percentage of Annexin V–positive RBCs.

**Quantification of pro-inflammatory cytokine levels in serum.** C57BL/6J mice (8–12 weeks, *n* = 3) were transfused with either fresh RBCs or RBCs stored for 17 days (untreated or intermittently treated with *Tau-Acr*NFS). After 2 h of transfusion, blood was collected and plasma was separated and analyzed for pro-inflammatory markers at 1:10 dilution with array buffer. Cytokines/chemokines, including stromal cell-derived factor1 (CXCL12), macrophage-colony-stimulating factor (M-CSF), complement component 5 A(C5A), and keratinocyte-derived chemokine/CXCL1 (KC/CXCL1) were quantified using Mouse Cytokine Array Panel A (R&D Systems, Minnesota, USA, Serial No. 893560) following the manufacturer's instruction.

### Transfusion of RBCs
**Biotinylation of RBCs, in vivo.** Briefly, Sulfo–NHS Biotin (Thermo-fisher Scientific, US) powder was dissolved in sterile PBS, vortexed, and passed through a 0.22 μm filter. It was diluted to a final concentration of 1 mg biotin–sulfo–NHS per 300 μL 1× PBS. Mice were restrained, and the biotin reagent solution was administered through intravenous injection. After 48 h of the biotin administration, under mild

anesthesia, around 1 ml of biotinylated blood was collected through a cardiac puncture in CPD-containing vials and immediately centrifuged at 500*g* for 20 min at 4 °C. Plasma was carefully aspirated, and pRBCs were pooled together. The biotin-labeled blood was processed and stored using Optisol as a preservative in sterile Eppendorf tubes (0.5 and 1 ml).

**Treatment of Biotin-labeled RBCs with Tau-AcrNFS Functionality tests.** Biotin-labeled RBCs were divided into treated and untreated groups. The treatment group (*n* = 3) pRBC units were treated with *Tau-Acr*NFS on the 5th and 10th days of storage and then packed until 18 days. The untreated group pRBCs units (*n* = 3) were not treated with sheets during the storage timeline. Both groups were analyzed for PS exposure during storage.

**Evaluating the enhanced shelf-life of biotin-labeled RBCs by recovery experiment.** On different days of storage (0, 7, 14, 15, 16, 17, and 18 days), biotinylated pRBC units with and without treatment were diluted (1:1) with PBS. Totally, 200 μl of diluted biotinylated pRBC was administered to the restrained mice via intravenous injection[45]. At specific time points, 6 μl of blood was collected via tail snip and stored in microfuge tubes containing 250 μL of sterile PBS.

To detect biotinylated RBCs through flow cytometry, approximately 10$^6$–10$^8$ RBCs (100 μL of the diluted sample) were added to 0.125 μg of streptavidin– APC-efluor® 780 Conjugate (1:20 of 0.2 mg/mL; eBioscience™). Samples were incubated in the dark for 5 min, washed with 400 μL PBS, and centrifuged at 1000 g for 4 min. The RBC pellet was resuspended in 200 μL PBS. Samples were analyzed using BD LSR Fortessa cytometer (BD biosciences, USA). The red laser was used to excite the dye (λ emission = 780 nm) with a bandpass filter of 780/60. The following formula was used to calculate the post-transfusion survival (%).

$$\frac{Post\,transfusion\,RBC\,recovery\,(\%)\,at\,t(t)}{Post\,transfusion\,RBC\,Recovery\,at\,t(1h)} \times 100$$

where (*t*) stands for pdst-transfusion time point, i.e., 1, 24, 48, and 72 h. This formula is used for all pRBCs units stored for different days and then transfused in a different set of mice. The data was collected with FACSDiva v8.02 and data analysis was performed FlowJo v10.0.8.

### Quantification of total iron in the spleen
**Sample digestion and preparation.** Splenectomy was performed in mice after 2 h of transfusion. Spleen from all the groups was taken in a tared 2 ml Eppendorf, and the dry weight was recorded. Then, 1 ml protein precipitation solution and 1 ml (8%) nitric acid were added to the spleen tube and homogenized for 2 min. The protein precipitation solution was prepared with 0.53 N HCl and 5.3% trichloroacetic acid in HPLC water. The solution containing a homogenized spleen with a closed lid was boiled at 2000C for 30 min. Cooled in RT for 2 min, and caps were opened to release air bubbles. Finally, the solution was centrifuged at full speed for 10 min. For total iron quantification, the supernatant from all the groups (*n* = 3) was taken.

**ICP–MS method.** The optimized operating conditions used for ICP–MS (Shimadzu-2030) are summarized in Supplementary Table 2. A peristaltic pump was used to deliver the samples (0.1 mL/min) to the nebulizer, converting the sample into a spray mist using argon (Ar) gas. Automated adjustments were made for torch alignment, detector voltage, and ion lens voltages for optimized resolution, sensitivity, and stability across a broad range of atomic masses. The doubly charged ion/charge ratio (Ce$^{2+}$ 69.95 to Ce 139.90) and oxide ratio (CeO$^+$ 155.90 to Ce139.90) were also monitored and were maintained below 3% and 2.5% (intensity), respectively. The Prep Fast system was interfaced with the ICP–MS and auto-diluted the stock solutions to generate the

calibration curves and QCs samples. The 30-ppb Indium (In) solution was also mixed in-line by the Prep-Fast during the analysis as an internal standard to all the analyzed samples and standards. The method was evaluated for its selectivity, sensitivity, linearity, precision, and accuracy.

### Statistical analysis

In experiments with multiple groups, ordinary one-way ANOVA with Tukey's post hoc test was used. In experiments with multiple groups, which are time-course studies, repeated measures of one-way ANOVA were used. The two-tailed Student's $t$-test with Welch's corrections was used to compare two experimental groups. The probability value ($P$) < 0.05 was considered as a statistically significant difference. Statistical analysis and graphing were performed with GraphPad PRISM 9.

### Reporting summary

Further information on research design is available in the Nature Portfolio Reporting Summary linked to this article.

## Data availability

All data generated or analyzed during this study are included in this published article and its supplementary information files or are available from the corresponding author upon reasonable request. Source data are provided with this paper.

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

## Acknowledgements

This work was supported by core funds from the Institute for Stem Cell Science and Regenerative Medicine (DBT-inStem). S.P. was supported by Senior Research Fellowship from the Lady Tata Memorial Trust. U.B. was supported by National Postdoctoral Fellowship under the Nanomission of the Department of Science Technology (DST), Govt. of India. Animal work in the inStem/NCBS Animal Care and Resource Centre was partially supported by the National Mouse Research Resource (NaMoR) grant (grant no. BT/PR5981/MED/31/181/2012;2013-2016;2018) from the Department of Biotechnology. We thank the animal house facility/ members and Central Imaging & Flow Cytometry facility (CIFF) at inStem and NCBS. We thank the NMR facility at NCBS. P.K.V. thanks Dr. Sandip Patil and E-Spin Nanotech Pvt. Ltd., India, for their generous gift of a Super ES-1 device.

## Author contributions

P.K.V. conceived and designed the experiments. S.P., M.M., P.S., U.B., and R.P. designed, performed the experiments, and analyzed the data. M.M., S.P., and T.J. synthesized the polymers. S.P., M.M., P.S., U.B., and R.P. performed in vitro and in vivo experiments. S.P. and P.K.V. wrote the paper, all authors discussed the results. P.K.V. supervised the project.

## Competing interests

P.K.V., S.P., M.M., P.S., and U.B. hold a patent related to this technology: "Composition and methods to enhance the quality and shelf-life of biological material" (IPP Application no: 202241014827, provisional application). The patent applicant is the Institute for Stem Cell Science and Regenerative Medicine. All compositions and methods of use described in this manuscript were covered in the patent application. T.J.G. and R.P. do not have any competing interests.
