## [Peer Review File · Nature Communications]

REVIEWER COMMENTS

Reviewer #1 (Remarks to the Author):

The manuscript entitled „Intermittent scavenging of storage lesion from stored red blood cells by electrospun nanofibrous sheets enhances the quality and shelf-life of stored RBCs" by Pandey et al. describe the development of a taurine and acridine-containing electrospun nanofiber fabric (Tau-AcrNFS) that could sequester DAMPs such as cell-free DNA, nucleosomes, free hemoglobin, and polyunsaturated fatty acids from stored red blood cells to improve their quality and shelf life.

The authors synthesized positively charged polyacridine and negatively charged polytaurine polymers and then generated nanofibers from the polymers by electrospinning. Intermittent contact of the stored red blood cells with a device coated with tau-AcrNFS effectively scavenged DAMPs. The "purification" process removes DAMPs from stored blood and improved red cell membrane integrity ex vivo, demonstrating improved quality of the stored red blood cells and longer shelf life in mouse transfusion studies.

The study seems innovative and novel but needs some improvements prior to publication.

1) First of all, the title of the manuscript is difficult to understand, I would propose to change the title in order to avoid repetitions ... "Intermittent scavenging of storage lesion from stored red blood cells by electrospun nanofibrous sheets enhances the quality and shelf-life of stored RBCs"

... maybe to something similar like:

"Interception of storage damage fragments from stored red blood cells by electrospun nanofibers helps to by improves their quality and shelf life"

2) The storage conditions for the red blood cells seem to be artificial. RBCs are stored in "20 ml vacutainers" in the present work - but nothing is written about the material of the vacutainers. Recognized state of the art for storing red blood cells are currently PVC bags that are oxygen diffusible and from which the plasticizer DEHP is dissolved out, which is most beneficial for the quality of the red blood cells - possibly by membrane plasticization (see e.g. DOI: 10.1097/SHK.0000000001646, <https://doi.org/10.1159/000490502>). Since the authors do not use these state-of-the-art systems, their rate of improvement is questionable or not linked to the correct reference. I would like to encourage the authors to compare their new method with RBCs stored in a medical device PVC bag containing the plasticizer DEHP. Therefore, at least the in vitro key experiments should be repeated with the correct reference conditions.

3) The conditions for the polymer leaching test with 30 + 5 min in double distilled water is not efficient enough. In order to really remove polymer fragments of the highly charged but hydrophobic polymers I would recommend to use at minimum salt solutions with similar or higher ionic strength compared to preservative composition used for the storage of the red blood cells in a leaching test (in order for electrostatic shielding). If the authors consider their product to become a medical device according to European CE regulations, leaching tests have to be performed at elevated temperatures in organic solvents as well (to mimic interactions with condensed blood cell solution). Leaching is in particular highly relevant due to the fact that the DEHP plasticizer problematic is present in all notified bodys/registration institutions.

4) The authors should comment on the feasibility of their technology/procedure in terms of translation. What does it mean logistically if the blood has to be pumped through a "filter" system from time to time? - That seems like a pretty tall order to me. Would permanent sequestration of DAMPS with an insert in a PVC bag be an alternative? This would be much more feasible to become a

medical device than to implement a new red blood cell storage procedure.

Reviewer #2 (Remarks to the Author):

the results / findings of the study are significant and data representation in form of schematics for specific markers and their depiction are excellent.

the findings will open new avenues towards usage of improved methods of blood storage. Methodology is in detail and well explained and are reproducible.

the conclusions are have been supported sufficiently with results and schematics and quantification data.

However a few suggestions towards minor revisions:

- Young RBCs (yRBCs) can be mentioned as fresh RBCs in the text similar to fig 1., to have more clarity, as it is in context with the storage days and not with reference to actual mixed population. i.e. aging.
- ‘Through RBCs transfusion studies in mice, we have demonstrated’ reframe this sentence so as to rectify the grammar.
- Reframe the sentence “Using poly-Tau and poly-Acr polymers, we have generated anionic (Tau-NFS) and cationic (Acr-NFS) electrospun nanofibrous sheets using the conventional electrospinning technique”
- Reframe the sentence ‘To conduct RBCs transfusion experiments, we have labelled RBCs with biotin’
- All the Figure legends should also explain the symbols of open circles and open triangles represented on the bars in the graphs.
- Supplementary figures should also have detailed legends.

Reviewer #3 (Remarks to the Author):

This manuscript describes an adsorption process using the authors’ ‘in-house’ synthesized ionic-charged polymer electrospun nanofibrous sheets (NFS) to deplete DAMPs (damage-associated molecular patterns), such as DNA, nucleosomes, cytokines and other bioactive cell breakdown products. Negative and positive ionic charged polymers were achieved by linking taurine and acridine, respectively. The manuscript focusses on the potential of the Tau-AcrNFS technology to improve the quality of RBC samples stored in conditions similar to those used for the storage of RBC concentrates for transfusion. The RBC samples were not specifically depleted of leukocytes or platelets before storage. The samples were treated midway through the storage period, i.e. at day 21 or day 28 of the total 42 days storage for human RBCs and, days 5 and 10 of the total 14 days storage time for mouse RBCs. Analyses included in vitro assessments and an in vivo mouse RBC transfusion model. The authors report significant improvement in the quality of Tau-AcrNFS-treated RBC samples compared to untreated RBC controls.

Major comments:

1. The idea of removing bioactive factors and metabolic biproducts from transfusion products is not new. Indeed, efforts have focussed more on prevention rather than remedial strategies, which is akin to the adage “closing the stable door after the horse has bolted”. The DAMPs that the authors measured are all associated, directly or indirectly, with the presence of leukocytes (and platelets) in the RBC samples; accumulation of these DAMPs could have been effectively prevented by removing the leukocytes and platelets prior to sample storage. The authors allude to this in the Introduction (P2, 2nd paragraph) by reference to leukocyte-depletion filtration of RBC concentrates, which is a routine process in many parts of the world where universal pre-storage leukocyte-reduction filtration

has been implemented. Although the authors are correct to say that RBC-derived 'DAMPs', such as cell-free Hb, continue to accumulate during storage of leukoreduced RBC concentrates, the levels are significantly reduced compared to non-leukoreduced RBC concentrates (e.g. Hess JR et al, *Transfusion* 2009;49:2599-2603). Knowing this, why did the authors use non-leukoreduced RBC samples for their experiments? In so doing significantly lessens the interest and relevance of their work to the target scientific audience.

2. The RBC samples were treated midway through the storage period. Interventions of this nature midway through storage of RBC transfusion products would not be feasible. Therefore, the authors experimental design is impractical from a 'real world' perspective. It is not clear how the Tau-AcrNFS technology could be applied in the setting of stored transfusion products. The authors need to provide commentary in the Discussion to address this point, especially in light of existing pre-storage leukocyte-reduction technology, as indicated in Comment #1.

3. The novelty of the work rests entirely on the authors' Tau-AcrNFS technology. Harnessing ionic charge as the basis of an adsorption matrix is not new. For example, the chemistries of leukocyte-reduction filters are based on ionic charge matrices. The authors need to provide more definitive proof of the innovative aspect of their technology.

4. Hemocompatibility.. The authors state: "The charged Tau-AcrNFS are hemo-compatible, do not cause hemolysis, and upon incubation, RBCs or any other cells do not adhere to these sheets, suggesting that Tau-AcrNFS are robust and consist of key parameters that play a critical role in translating this technology into a medical device" (Discussion, P9, 1st paragraph, last sentence). No mention is made of assessment for leaching of chemical entities from the Tau-AcrNFS polymers into the RBC supernatant. Since acridine is an irritant and can be carcinogenic, proof of biosafety is a critical consideration. The authors should include further data if available, and discussion of their rationale in choosing acridine rather than a 'safer' cation.

Note: this reviewer acknowledges that the authors conducted polymer leaching experiments as part of the initial work-up (Results P4; Methods, P13), but these experiments were conducted in water, which is not comparable to incubation in a suspension of blood cells.

5. It is not clear how the RBC samples were exposed to the 8 cm² Tau-AcrNFS. Were the sheets directly inserted into the storage tubes? How were the mouse RBC samples treated given that they were much smaller volumes/smaller tubes?

6. Human RBC samples were treated only once (i.e. day 21 or day 28), but the mouse RBCs used in the in vivo transfusion experiments were treated twice (i.e. day 5 and day 10). Why the difference in treatment strategy? What impact does repeat treatments have compared to the single treatment?

7. Other characteristics of the RBC storage lesion include leakage of K⁺ ions and shedding of extracellular vesicles (EVs) – the latter having immunomodulatory and procoagulant potential. Is the Tau-AcrNFS technology effective in depleting these types of RBC-derived 'DAMPs'?

8. Taking a purist view, DAMPs refers to endogenous molecules released from damaged or dying cells and that can activate the innate immune system via interaction with pattern recognition receptors. The array of physicochemical changes that occur to RBCs, encapsulated by the term 'RBC storage lesion', are considerably more diverse than the release of DAMPs. Indeed, oxidative stress is the hallmark of the RBC storage lesion – a point that seems to have been overlooked. Statements that infer that the 'RBC storage lesion' and 'DAMPs' are equivalent in meaning (e.g. Abstract, 1st sentence: "...upon storing RBCs, a wide range of storage lesion, also known as damage-associate molecular patterns (DAMPs)....") are misleading and need to be revised. Other statements, such as "In situ generated storage lesion, DAMPs are the primary cause of declining RBCs condition" (Discussion, 2nd sentence), are also inaccurate. A more descriptive and balanced account of the RBC storage lesion

needs to be provided in the Introduction.

Specific comments

Introduction

P2, 2nd paragraph. Washing stored RBCs to remove accumulated bioactive factors is another well-established strategy approved for use in transfusion medicine. The authors should mention this.

Results

P4, 1st paragraph. 1) The SEM images of the Tau-AcrNFS after incubation with RBC samples (Fig 1e) look quite different to the before treatment images (Fig 1b) and suggest that there is significant gumming of the Tau-AcrNFS. Have the authors evaluated the total protein and lipid profiles of the RBC supernatant following Tau-AcrNFS treatment to determine what other proteins/lipids are removed in addition to the measured DAMPs? 2) Author statement: "Scanning electron microscope images of Tau-Tau-AcrNFS and Acr-Tau-AcrNFS post-incubation with stored RBCs show that RBCs or neutrophils do not adhere to Tau-AcrNFS (Fig. 1e); hence, there will not be any loss of RBC count due to the Tau-AcrNFS treatment." Do the authors have RBC counts pre- and post- treatment to support their claim that there is no loss of RBCs? If not, this claim should be revised 3) Author statement: "A <1% and <3% of hemolysis was observed for Tau-Tau-AcrNFS and Acr-Tau-AcrNFS, respectively (Fig. 1f), which were below than acceptable range (~5%)." What does the "acceptable range (~5%)" refer to?

P4, 3rd paragraph. Of the treatment conditions tested, the smallest Tau-AcrNFS area size (8 cm²) and shortest incubation time (5 mins) were found to be as effective as the larger/longer treatment conditions tested. In other words, the minimum limits for treatment conditions were not identified. Do the authors think that smaller area size/shorter treatment time could be as effective?

P5, 2nd paragraph. Human RBC samples were treated on either day 21 or day 28 based on the rationale that this was the point at which the level of DAMPs increased exponentially (Fig 2a-f). Nevertheless, in the 'real world' the median storage duration of RBC concentrates is typically <21 days as reported in several large multicenter 'age of blood' randomized clinical trials, such as ABLE, RECESS, TRANSFUSE. It would be relevant for the authors to add comment in the Discussion.

P5 3rd paragraph. RBC morphology. 1) Early stages of echinocytic shape change during in vitro storage are reversible once the RBCs are transfused. Only counting the number of discocytes would be an under-estimate of the proportion of stored RBCs that are likely to be functional in vivo. The SEM images in Fig 4b show the presence of echinocyte I and II shapes in all three samples of day 42 stored RBCs (untreated, 21 day treated and 28 day treated). The authors could review their morphology counts to take this into account. 2) Did the authors assess the morphology of the RBCs immediately before and sooner after treatment, e.g. 24 hrs afterwards? It would be interesting to know whether the morphology is significantly altered by the treatment.

P6, last paragraph. The FDA's requirement that >75% of transfused RBCs must be recoverable in the in vivo circulation 24 hrs after transfusion applies to human RBCs only. This point should be clarified here and elsewhere in the manuscript. To make comparisons with the recovery of transfused mouse RBC is irrelevant.

P8, 2nd paragraph, last sentence. Statement: "Therefore, three days of enhanced shelf-life in murine RBCs is equivalent to nine days in human RBCs." There is no robust science to support this extrapolation. The statement needs to be revised or deleted.

P8, 3rd paragraph. Mouse cytokine levels. 1) The R&D Systems mouse cytokine array tests for 40 different cytokines, chemokines, and acute phase proteins. The results for 4 bioactive factors are shown in Fig 6h. The authors should include comment about the other factors, i.e. were no differences found? 2) Ideally, images of representative arrays should be provided – as supplementary information. 3) Mouse blood samples were collected 2hrs after transfusion (Methods P17). The authors

should make this clear in the Results section, otherwise the reader would assume it was post-24 hrs.

Discussion

P9, 2nd paragraph, sentence: "When Tau-AcrNFS treatment is used terminally,.....". By "terminally", do the authors mean at the end of RBC storage shelf-life? Clarification is needed.

P9, 2nd last line. Instead of "phenomenon", suggest change to "hypothesis" or "expectation".

P10, 3rd line. Author statement: "Systemic inflammation is one of the hallmarks of transfusion of stored oldRBCs". This may be the case for transfusion of non-leukoreduced stored RBCs, but does not hold up for pre-storage leukoreduced stored RBCs. Furthermore, immunosuppression, rather than inflammation, has been documented in some clinical settings. Instead of "systemic inflammation" the authors would be advised to use the term "immunomodulation".

Methods

P13. Hemolysis assay. The equation used to calculate the % hemolysis used a ratio of the OD readings of the sample supernatant and the 100% lysis control. This is not valid as it assumes a linear relationship between the free Hb concentration and the OD reading, which is not necessarily true. The authors need to revisit these data.

P19. At what time-point post transfusion were the splenectomies performed on the mice?

Other comments

- "Intermittent treatment" gives the impression of randomness, whereas the Tau-ArcNFS treatments were performed at selected time points during storage of the RBC samples. An alternative term may be better.
- Use of the verb "prevented" needs to be reviewed throughout. In most instances, the effect was not prevented per se, but rather diminished or lessened.
- Some of the wording could be tightened. There are instances throughout where sentences do not make sense and need to be rephrased. For example, P7, Section titled 'Intermittent scavenging of storage lesion with Tau-AcrNFS enhances the shelf-life of stored RBCs' has several sentences that need to be rephrased.
- Statistical analysis of time course experiments in which the same sample is tested at multiple different time-points, such as for the data in Fig 2a-f and Fig 5a-b, should use repeated measures ANOVA, not one-way ANOVA with Tukey's post-hoc.

Reviewer #4 (Remarks to the Author):

The creating of anionic and cationic electrospun fibers with said additives is not novel, examples are previously published. The application of such fibers is novel.

Was the degree of charges created optimized, or amounts added to functionalize?

What is the loading capacity of scavenged contents per unit fiber surface area? How do you know the fibers are scavenging the components described? How much fouling of charges occurs with other components such as plasma proteins, decreasing potential efficiency? Is this "other" absorption key or detriment?

In the introduction the authors state that "technologies to prevent the formation of DAMPs in stored

RBCs remains an unmet need." This proposed technology is not prevent DAMPs but for chelating them to prevent associated damage.

More studies are needed to show above and that scavenging is occurring and determining the loading capacity.

We thank all the reviewers and the editor for their positive responses and for finding our work innovative, novel, and of potential interest. The reviewers and editors have decided to invite a major revision of the manuscript. Thus, we have revised the manuscript and performed the suggested experiments to answer the reviewers' comments. Please see below for our response to each point that the reviewers have raised.

Reviewer # 1. We thank the reviewer for critically reading our manuscript and for appreciating the novelty of this work. We thank her/him for providing constructive feedback. Please see below for our response to each point that has been raised by the referee.

Comment R1-1: First of all, the title of the manuscript is difficult to understand, I would propose to change the title in order to avoid repetitions ... "Intermittent scavenging of storage lesion from stored red blood cells by electrospun nanofibrous sheets enhances the quality and shelf-life of stored RBCs"

... maybe to something similar like: "Interception of storage damage fragments from stored red blood cells by electrospun nanofibers helps to by improves their quality and shelf life"

Reply: We thank the reviewer for suggesting a new title to avoid repetitions of words. Hence, as suggested, we have modified the title by removing repetitive words like 'stored RBCs'. The revised title is "Intermittent scavenging of storage lesion from stored red blood cells by electrospun nanofibrous sheets enhances their quality and shelf-life."

Comment R1-2: The storage conditions for the red blood cells seem to be artificial. RBCs are stored in "20 ml vacutainers" in the present work - but nothing is written about the material of the vacutainers. Recognized state of the art for storing red blood cells are currently PVC bags that are oxygen diffusible and from which the plasticizer DEHP is dissolved out, which is most beneficial for the quality of the red blood cells - possibly by membrane plasticization (see e.g. Melzak K. A. et al, *Transfus. Med. Hemother.* 2018;45:413-422)). Since the authors do not use these state-of-the-art systems, their rate of improvement is questionable or not linked to the correct reference. I would like to encourage the authors to compare their new method with RBCs stored in a medical device PVC bag containing the plasticizer DEHP. Therefore, at least the in vitro key experiments should be repeated with the correct reference conditions.

Reply: We thank the reviewer for pointing out this. We apologize for the oversight while writing this methods part. The "20 ml vacutainers" were used only while collecting blood, but stored in standard bags. In this study, we have stored human RBCs and mouse RBCs. For storing human RBCs, we have used standard recommended PVC storage bags and obeyed the appropriate state-of-the-art system as recommended NACO & FDA guidelines (<http://naco.gov.in/sites>) (<https://www.fda.gov/vaccines-blood-biologics/biologics-guidances/blood-guidances>). Furthermore, mouse RBCs were not stored in storage bags, instead, due to small volumes, they were stored in standard Eppendorf tubes containing CPD-A, and sufficient residual air space was kept. The Eppendorf tubes were stored in the dark at 4 °C. This is a standard practice in the field, earlier this process was adopted to store both human and mouse RBCs (Gilson C. R. et al, *Transfusion* 2009;49:1546-1553). The quality control to test the acceptable storage condition is; hemolysis of stored RBC units containing CPD-A should not exceed 1%, and post-transfusion of RBCs *in vivo*, more than 70% of transfused

RBCs should be recovered after 24 hr of transfusion. The mouse RBCs, which we have stored, have shown both these parameters. Accordingly, we have revised the text in the revised manuscript.

Comment R1-3: The conditions for the polymer leaching test with 30+5 min in double distilled water is not efficient enough. In order to really remove polymer fragments of the highly charged but hydrophobic polymers I would recommend to use at minimum salt solutions with similar or higher ionic strength compared to preservative composition used for the storage of the red blood cells in a leaching test (in order for electrostatic shielding). If the authors consider their product to become a medical device according to European CE regulations, leaching tests have to be performed at elevated temperatures in organic solvents as well (to mimic interactions with condensed blood cell solution). Leaching is in particular highly relevant due to the fact that the DEHP plasticizer problematic is present in all notified body's/registration institutions.

Reply: We thank the reviewer for this critical suggestion. As the reviewer suggested, we now performed a leaching test in Optisol (AS-5) solution [composition: sodium chloride (8.77 mg/mL), Glucose anhydrous (8.18 mg/mL), Mannitol (0.01% w/v) and Adenine (0.03 mg/mL)]. Optisol is used as a standard preservative for storing red blood cells. To demonstrate this, in independent experiments, *Acr*-NFS (8 cm², 6.75 mg of *poly*-Acridine) and *Tau*-NFS (8 cm², 14.42 mg of *poly*-Taurine) were incubated in 1 ml of Optisol solution for 2 hrs at 4 °C. Although in scavenging experiments we incubate nanofibrous sheets for only 15 mins, for a safer side, in a leaching test we incubated for 2 hours. Post 2-hour incubation, leached polymers in Optisol solution were quantified as mentioned in the Methods section. Interestingly, under these high salt conditions as well, the quantification of leached polymers shows that neither polymers have any significant leaching (<0.2%). Accordingly, we have modified the revised manuscript.

Comment R1-4: The authors should comment on the feasibility of their technology/procedure in terms of translation. What does it mean logistically if the blood has to be pumped through a "filter" system from time to time? - That seems like a pretty tall order to me. Would permanent sequestration of DAMPS with an insert in a PVC bag be an alternative? This would be much more feasible to become a medical device than to implement a new red blood cell storage procedure.

Reply: We thank the reviewer for critical feedback on the aspect of translating our technology as a medical device. We second reviewer's thoughts on making this technology as a medical device instead of a storage bag. We are brainstorming to have a final 'product design' to translate the technology. In this work, as a proof-of-concept we establish that intermittently scavenging of DAMPs using *Tau-Acr*NFS can enhance the quality and shelf-life. Now, we are focusing on designing the clinical product. We do not want to use this technology as a filtration device, which is cumbersome and logistically not feasible, but as the reviewer mentioned, we are designing an insert in PVC bag that could scavenge DAMPS and could be removed. In the next product development stage, it will be finalized.

Reviewer # 2. We thank the reviewer for finding the results of our work significant and appreciating the schematics used for data representation. Please see below for our response to each point that has been raised by the referee.

Comment R2-1: Young RBCs (yRBCs) can be mentioned as fresh RBCs in the text similar to fig 1., to have more clarity, as it is in context with the storage days and not with reference to actual mixed population. i.e., aging.

Reply: As reviewer suggested, young RBCs (yRBCs) were changed to fresh RBCs to keep it consistent with entire manuscript. Accordingly, it has been revised in the manuscript.

Comment R2-2: “Through RBCs transfusion studies in mice, we have demonstrated” reframe this sentence so as to rectify the grammar.

Reply: We thank the reviewer for pointing out the grammatical error. We have revised the text to rectify the grammar.

Comment R2-3: Reframe the sentence “Using poly-Tau and poly-Acr polymers, we have generated anionic (Tau-NFS) and cationic (Acr-NFS) electrospun nanofibrous sheets using the conventional electrospinning technique.

Reply: We have rectified the grammatical error in the revised manuscript.

Comment R2-4: Reframe the sentence “To conduct RBCs transfusion experiments, we have labelled RBCs with biotin”

Reply: We have rectified and reframed the sentence. Additionally, we carefully gone through the manuscript to avoid similar writing style, and corrected them.

Comment R2-5: All the Figure legends should also explain the symbols of open circles and open triangles represented on the bars in the graphs.

Reply: We understand the confusion caused by open circles and open triangles. The groups are clearly labelled on the figures. Therefore, to avoid any confusion, we have revised the figures, and kept only open circles to represent all data points.

Comment R2-6: Supplementary figures should also have detailed legends

Reply: We thank the reviewer for this suggestion. Based on this suggestion, we have included detailed figure legends for all supplementary figures in the revised version.

Reviewer # 3. We thank the reviewer for critically reading our manuscript and providing their constructive feedback. Please see below for our response to each point that has been raised by the referee.

Comment R3-1: The idea of removing bioactive factors and metabolic biproducts from transfusion products is not new. Indeed, efforts have focused more on prevention rather than remedial strategies, which is akin to the adage “closing the stable door after the horse has bolted”. The DAMPs that the authors measured are all associated, directly or indirectly, with the presence of leukocytes (and platelets) in the RBC samples; accumulation of these DAMPs could have been effectively prevented by removing the leukocytes and platelets prior to sample storage. The authors allude to this in the Introduction (P2, 2nd paragraph) by reference to leukocyte-depletion filtration of RBC concentrates, which is a routine process in many parts of the world where universal pre-storage leukocyte-reduction filtration has been implemented. Although the authors are correct to say that RBC-derived ‘DAMPs’, such as cell-free Hb, continue to accumulate during storage of leukoreduced RBC concentrates, the levels are significantly reduced compared to non-leukoreduced RBC concentrates (e.g., Hess JR et al, *Transfusion* 2009;49:2599-2603). Knowing this, why did the authors use non-leukoreduced RBC samples for their experiments? In so doing significantly lessens the interest and relevance of their work to the target scientific audience.

Reply: We thank the reviewer for discussing this critical point. Due to two reasons, we have performed experiments with non-leukoreduced RBCs.

One of the primary reasons is the cost. The leukoreduction process is expensive; hence, most of the blood banks in India and developing countries do not use leukoreduction processes routinely (Sharma R. R. et al, *Asian J. Transfus. Sci.* 2010;4:3-8). Thus, the cost implicated in universal leukoreduction policy is not practically feasible, especially in developing countries and other under-resourced nations. In fact, before initiating the project, we interacted with several blood banks across India to get their feedback. Surprisingly, only <10% of blood banks use the leukoreduction process. Only in the case where recipient is immunosuppressed/immuno-compromised, in such cases, they use leukoreduced blood. Therefore, if our technology can make non-leukoreduced blood safe for transfusion, it will have a major impact.

Additionally, we agree with the reviewer about the significance of leukoreduced blood usage in the clinic to reduce the incidence of blood transfusion complexities. It is observed that leukoreduced blood units have lesser WBCs associated lesions like extracellular DNA and free histone by the end of the storage period (Fuchs T. A. et al, *Transfusion* 2013;53:3210-3216). However, several poly-unsaturated fatty acids (PUFAs) and oxidized products are still accumulated in leukoreduced blood units akin to Non-Leukoreduced blood units. Indeed, PUFAs and their oxidized products are known to cause transfusion-related acute lung injury (TRALI) (Silliman C. C. et al, *Transfusion* 2011, 51:2549-2554). Therefore, we have also quantified the DAMPs in leukoreduced stored RBCs. The data suggest that DAMPs such as free Hb and PUFAs are significantly produced at 42 days (See **Fig. R1**, only for review). We have also demonstrated that upon transfusion of leukoreduced stored oldRBCs in mice elicited significant systemic inflammation. Newly developed *Tau-Acr*NFS could scavenge DAMPs from leukoreduced blood and reduce transfusion-related systemic inflammation (data not

shown). We are extensively studying with leukoreduced stored RBCs, which will be communicated for publication shortly.

Fig. R1. Hypothermal storage of human leukoreduced (LR) RBCs induced DAMPs production, and charged *Tau-AcrNFS* efficiently scavenged DAMPs from stored LR RBCs. a-e, Storage-induced production of RBCs associated DAMPs as a function of time. LR-RBCs stored at 4°C progressively produce free hemoglobin (Hb, **a**), and polyunsaturated fatty acids (PUFAs) such as AA (**b**), 5-HETE (**c**), 12-HETE (**d**), and 15-HETE (**e**). **f-j,** Incubation of DAMPs accumulated 42 days-stored LR-RBCs with *Tau-AcrNFS*, for 5 mins at 4 °C efficiently scavenged and reduced the concentration of DAMPs significantly. The relative concentration of DAMPs before and after *Tau-AcrNFS* treatment has significantly reduced accumulated Hb (**f**), and PUFAs (**g-j**). Data are mean ± s.d. (n = 3, performed at least twice). For **a-e**, *P* values were determined by using Repeated Measures ANOVA, and for **f-j**, by Student’s *t*-test with Welch’s correction using GraphPad PRISM 9, and exact *P* values are indicated.

Comment R3-2: The RBC samples were treated midway through the storage period. Interventions of this nature midway through storage of RBC transfusion products would not be feasible. Therefore, the authors experimental design is impractical from a ‘real world’ perspective. It is not clear how the Tau-AcrNFS technology could be applied in the setting of stored transfusion products. The authors need to provide commentary in the Discussion to address this point, especially in light of existing pre-storage leukocyte-reduction technology, as indicated in Comment#1.

Reply: We understand the reviewer’s concern about the mode of use in practical scenarios. We are considering making this technology as a medical device instead of a storage bag. We are brainstorming to have a final ‘product design’ to translate the technology. In this work, as a proof-of-concept, we establish that intermittently scavenging DAMPs using *Tau-AcrNFS* can enhance the quality and shelf-life. Now, we are focusing on designing a clinical product. We do not want to use this technology as a filtration device where intermittent filtration is required, which will be cumbersome and logistically not feasible. Therefore, we are designing an insert in a PVC bag that could scavenge DAMPs and could be removed. In the next product development stage, it will be finalized.

As the reviewer suggested, we have added a commentary in the Discussion of the revised manuscript.

Comment R3-3: The novelty of the work rests entirely on the authors’ Tau-AcrNFS technology. Harnessing ionic charge as the basis of an adsorption matrix is not new. For example, the chemistries of leukocyte-reduction filters are based on ionic charge matrices. The authors need to provide more definitive proof of the innovative aspect of their technology.

Reply: The current work has two novelty factors. 1). Demonstrating the concept of intermittent scavenging of DAMPs during storage could slow the aging of RBCs to enhance the quality and shelf-life of stored oldRBCs. 2) Material design to efficiently scavenge DAMPs through ionic interactions and intercalation process.

As the reviewer stated, the primary mechanism of leukocyte removal is the charge-based adhesion of negatively charged leukocytes to the filter material by Van der Waals and electrostatic forces (Sharma R. R. et al, *Asian J. Transfus. Sci.* 2010;4:3-8). However, in the leukoreduction process, size-based filtration of leukocytes is the primary mechanism. Additionally, cationic charge interaction was used to enhance the binding of leukocytes to the filters. On the contrary, in this work, electrostatic interactions were designed to bind only DAMPs and not any type of cells. For example, we performed a Complete Blood Count (CBC) before and after treatment with *Tau-AcrNFS*. Quantifying the number of RBCs, WBCs, platelets, neutrophils and lymphocytes before and after treatment with *Tau-AcrNFS* revealed that there is no loss of cells (**Fig. R2**), suggesting that cells do not adhere to these nanofibrous sheets.

This data now is included in the revised supplementary material (**Supplementary Figure 2**), and the text has been modified in the revised manuscript.

Fig. R2. A complete blood count of stored RBCs before and after *Tau-AcrNFS* treatment. a-e, Treatment of stored blood with *Tau-AcrNFS* did not cause loss of cells. Quantification of RBCs (a), WBCs (b), neutrophils (c), lymphocytes (d), and platelets (e) before and after treatment with *Tau-AcrNFS*, Data are mean \pm s.d. ($n = 3$). For a-e, P values were determined by Student's t-test with Welch's correction using GraphPad PRISM 9, ns = not significant.

Additionally, nanofibrous sheets were systematically designed to bind efficiently to extracellular DNA not only through electrostatic interactions but through intercalation as well. A combination of electrostatic interactions and intercalation enhances the DNA scavenging capacity. For example, we have prepared four nanofibrous sheets with four different polymers, poly-tetradecane, poly-pyridinium, poly-YO, and poly-acridine (Fig. R3a). While poly-tetradecane is a neutral polymer, poly-pyridinium, poly-YO and poly-acridine have a cationic charge. However, poly-pyridinium does not intercalate with DNA. On the contrary, YO and acridine are DNA intercalators. However, acridine is a stronger intercalator than YO (Kulyk O. G. et al, *Dyes and Pigments* 2022;200:110148 and Larsson A. et al, *J. Am. Chem. Soc.* 1994;116:8459-8465). Furthermore, we have prepared nanofibrous sheets using these polymers, subsequently DNA scavenging studies were carried out with 42 days stored oldRBCs. The relative DNA scavenging efficiency has shown in Fig. R3b. The data in Fig. R3b shows that scavenging efficacy is in the following order, *Tetradecane-NFS* < *Pyr-NFS* <

YO-NFS < Acr-NFS. These results clearly suggest that electrostatic interactions and intercalation play a key role in DNA scavenging.

Therefore, the current approach does provide the novelty in developing efficient technology to enhance the quality and shelf-life of stored RBCs.

Fig. R3. Ionic interactions and intercalation enhance DNA scavenging efficiency. a, Chemical structures of synthesized polymers that were used to generate nanofibrous scaffolds. **b,** Relative DNA presence in stored oldRBCs before scavenging and after scavenging with various nanofibrous scaffolds. For **b,** *P* values were determined by ordinary one-way ANOVA with Tukey’s post hoc analysis using GraphPad PRISM 9, and exact *P* values are indicated, ns = not significant.

Comment R3-4: Hemocompatibility. The author’s state: “The charged Tau-AcrNFS are hemo-compatible, do not cause hemolysis, and upon incubation, RBCs or any other cells do not adhere to these sheets, suggesting that Tau-AcrNFS are robust and consist of key parameters that play a critical role in translating this technology into a medical device” (Discussion, P9, 1st paragraph, last sentence). No mention is made of assessment for leaching of chemical entities from the Tau-AcrNFS polymers into the RBC supernatant. Since acridine is an irritant and can be carcinogenic, proof of biosafety is a critical consideration. The authors should include further data if available, and discussion of their rationale in choosing acridine rather than a “safer” cation

Note: this reviewer acknowledges that the authors conducted polymer leaching experiments as part of the initial work-up (Results P4; Methods, P13), but these experiments were conducted in water, which is not comparable to incubation in a suspension of blood cells.

Reply: We thank the reviewer for these critical points. To demonstrate the hemocompatibility of *Tau-Acr*NFS, we performed a Complete Blood Count (CBC) before and after treatment with *Tau-Acr*NFS. Quantifying the number of RBCs, WBCs, platelets, neutrophils and lymphocytes before and after treating with *Tau-Acr*NFS revealed that there is no loss of cells suggesting that these nanofibrous sheets do not cause any cell death or cell capture (**Fig. R2**, data is given in response to comment R3-3, reviewer-3). Furthermore, there was no reduction in RBC numbers, suggesting that *Tau-Acr*NFS treatment does not cause any hemolysis.

The reviewer's other point: The rationale for selecting poly-acridine instead of other safer cations is the following. As shown in the data in response to comment-3 (**Fig. R3**), the cumulative property of ionic interactions and efficient DNA intercalating property of poly-acridine makes it more attractive to generate efficient nanofibrous sheets. However, in the future, we will look for another safer cationic ligand.

As the reviewer pointed out, we have carried out polymer leaching studies in the water. Therefore, we now performed a leaching test in Optisol (AS-5) [composition: sodium chloride (8.77 mg/mL), Glucose anhydrous (8.18 mg/mL), Mannitol (0.01% w/v) and Adenine (0.03 mg/mL)]. Optisol is used as a preservative for storing RBCs. To demonstrate this, in independent experiments, *Acr*-NFS (8 cm², 6.75 mg of *poly*-Acridine) and *Tau*-NFS (8 cm², 14.42 mg of *poly*-Taurine) were incubated in 1 ml of Optisol solution for 2 hrs at 4 °C. Although in scavenging experiments nanofibrous sheets were incubated for only 15 mins, for a safer side, in leaching test nanofibrous sheets were incubated for 2 hours. Post 2-hour incubation, leached polymers in Optisol solution were quantified as mentioned in the Methods section. Interestingly, under these high salt conditions as well, the quantification of leached polymers show that neither polymers have any significant leaching (<0.2%). Accordingly, we have modified the revised manuscript.

Comment R3-5: It is not clear how the RBC samples were exposed to the 8 cm² *Tau-Acr*NFS. Were the sheets directly inserted into the storage tubes? How were the mouse RBC samples treated given that they were much smaller volumes/smaller tubes?

Reply: We apologize for the confusion. The scavenging experiments were performed on ice in the aseptic condition. In our studies, *Tau-Acr*NFS was mounted on the chamber slides, and the blood samples (human or mouse) were added. It was made sure that the blood should cover the entire surface area and be in contact with nanofibers.

Comment R3-6: Human RBC samples were treated only once (i.e., day 21 or day 28), but the mouse RBCs used in the *in vivo* transfusion experiments were treated twice (i.e., day 5 and day 10). Why the difference in treatment strategy? What impact does repeat treatments have compared to the single treatment.

Reply: We thank the reviewer for pointing out this missing lead in the manuscript. For the RBC recovery experiment we wanted to have the best condition possible. Single time

treatment and twice treatment does have the effect on total DAMPs production. For example, in mouse RBCs, total DAMPs on 14th day were significantly lower in samples that were treated twice on 5th and 10th day, compared to samples treated once on either 5th or 10th day (**Fig. R4**). Therefore, for RBC recovery experiments, we have adopted this approach. We thank the reviewer for this query. This data in **Fig. R4** is now included in the revised supplementary material (**Supplementary Figure 4**), and accordingly the text is modified in the results section and discussion section in the revised manuscript.

Fig. R4. Single and twice intermittent scavenging of DAMPs using *Tau-AcrNFS* from stored RBCs can reduce the accumulation of DAMPs at 14 days of stored mice RBCs. Intermittent treatment of stored RBCs twice with *Tau-AcrNFS* on 5th and 10th day significantly reduced the accumulation of DNA (**a**), and Hb (**b**) compared to the samples that were treated only once either on 5th day or 10th day. Data are mean \pm s.d. ($n = 3$, performed at least twice); P values were determined by ordinary one-way ANOVA with Tukey's post hoc analysis using GraphPad PRISM 9, and exact P values are indicated, ns = not significant.

Comment R3-7: Other characteristics of the RBC storage lesion include leakage of K^+ ions and shedding of extracellular vesicles (EVs) – the latter having immunomodulatory and procoagulant potential. Is the *Tau-AcrNFS* technology effective in depleting these types of RBC-derived ‘DAMPs’?

Reply: In this current study, we did not measure the production and scavenging of K^+ ions and extracellular vesicles (EVs). Although we did not quantify them, we anticipate that *Tau-AcrNFS* would have the potential to deplete K^+ ions and EVs, as we have observed efficient depletion of Fe^+ and a wide range of lipids (PUFAs), respectively. Additionally, post-transfusion of oldRBCs treated with *Tau-AcrNFS* did not elicit immunomodulation. Therefore, it could be possible that *Tau-AcrNFS* have depleted EVs as well. However, we are currently studying the efficiency of *Tau-AcrNFS* to deplete DAMPs from leukoreduced RBCs. In this study, we will also quantify the depletion of K^+ ions and EVs.

Comment R3-8: Taking a purist view, DAMPs refer to endogenous molecules released from damaged or dying cells and that can activate the innate immune system via interaction with pattern recognition receptors. The array of physico-biochemical changes that occur to RBCs, encapsulated by the term ‘RBC storage lesion’, are considerably more diverse than the release of DAMPs. Indeed, oxidative stress is the hallmark of the RBC storage lesion – a point that seems to have been overlooked. Statements that infer that the ‘RBC storage lesion’ and ‘DAMPs’ are equivalent in meaning (e.g. Abstract, 1st sentence: “...upon storing RBCs, a wide range of storage lesion, also known as damage-associated molecular patterns (DAMPs)...”) are misleading and need to be revised. Other statements, such as “In situ generated storage lesion, DAMPs are the primary cause of declining RBCs condition” (Discussion, 2nd sentence), are also inaccurate. A more descriptive and balanced account of the RBC storage lesion needs to be provided in the Introduction.

Reply: We thank the reviewer for this critical suggestion. As reviewer suggested, we have carefully gone through the text, and revised the text throughout the manuscript to reflect above suggestions.

Specific comments;

Comment R3-9: Introduction P2, 2nd paragraph. Washing stored RBCs to remove accumulated bioactive factors is another well-established strategy approved for use in transfusion medicine. The authors should mention this.

Reply: We thank the reviewer for pointing out this strategy. We have added the description of the same in the revised manuscript and appropriate references have been included.

References:

Schmidt, A., et al. "Proven and potential clinical benefits of washing red blood cells before transfusion: current perspectives." *International Journal of Clinical Transfusion Medicine* 2016;4:79-88.

Bennett-Guerrero, E. et al. "Randomized study of washing 40-to 42-day-stored red blood cells." *Transfusion* 2014;54:2544-2552.

Comment R3-10: Results P4, 1st paragraph. **1) The SEM images** of the Tau-AcrNFS after incubation with RBC samples (Fig 1e) look quite different to the before treatment images (Fig 1b) and suggest that there is significant gumming of the Tau-AcrNFS. Have the authors evaluated the total protein and lipid profiles of the RBC supernatant following Tau-AcrNFS treatment to determine what other proteins/lipids are removed in addition to the measured DAMPs?

2) Author statement: “Scanning electron microscope images of Tau-AcrNFS and Acr-Tau-AcrNFS post-incubation with stored RBCs show that RBCs or neutrophils do not adhere to Tau-AcrNFS (Fig. 1e); hence, there will not be any loss of RBC count due to the Tau-AcrNFS

treatment.” Do the authors have RBC counts pre- and post- treatment to support their claim that there is no loss of RBCs? If not, this claim should be revised.

3) Author statement: “A <1% and <3% of hemolysis was observed for Tau-NFS and Acr-NFS, respectively (Fig. 1f), which were below than acceptable range (~5%).” What does the “acceptable range (~5%)” refer to?

Reply: We thank the reviewer for these critical points.

To answer the first point; The observed gumminess of *Tau-Acr*NFS could be due to the accumulation of proteins and lipids on the nanofibrous sheets. Since we did not have any optimized method to extract proteins and lipids accumulated on the sheets, we have quantified the total proteins and lipids in the 42 days stored old RBCs before and after treatment with *Tau-Acr*NFS. As shown in **Fig. 2j-m** (main manuscript), bioactive lipids such as poly unsaturated fatty acid lipids have been significantly scavenged by *Tau-Acr*NFS. Similarly, post treatment with *Tau-Acr*NFS the total protein concentration also significantly reduced (**Fig. R5**). Therefore, the observed gumminess/smooth layer in the SEM could be due to the accumulation of proteins and lipids on the nanofibrous sheets. A sentence has been added in the revised manuscript to emphasize this point.

Fig. R5. Sequester of total protein using *Tau-Acr*NFS from 42 days stored oldRBCs.

Total protein content in the supernatant from 42 days stored oldRBCs was quantified before and after treatment with *Tau-Acr*NFS. A significant reduction in total protein content suggesting that proteins could be adhered to nanofibrous sheets. Data are mean \pm s.d. ($n = 3$); P values were determined by Student’s t-test using GraphPad PRISM 9, and exact P value is indicated.

To answer the second point; In addition to SEM images, we performed a complete blood count (CBC). Total number of cells (RBCs, WBCs, platelets, neutrophils and lymphocytes) were quantified before and after treatment with *Tau-Acr*NFS. The data given in **Fig. R2** (in response to comment R3-3, reviewer 3) suggest that there is no cell loss due the treatment. This data now is included in the revised supplementary material (**Supplementary Figure 2**), and the text has been modified in the revised manuscript.

To answer the third point; According to the international standard ASTM F756-17, a haemolytic index of >5% is classified as “hemolytic”, and <5% considered as hemocompatible (Thorarinsdottir, H. et al. *Sci. Rep.* 2022;12:8600).

Comment R3-11: P4, 3rd paragraph. Of the treatment conditions tested, the smallest Tau-AcrNFS area size (8 cm²) and shortest incubation time (5 mins) were found to be as effective as the larger/longer treatment conditions tested. In other words, the minimum limits for treatment conditions were not identified. Do the authors think that smaller area size/shorter treatment time could be as effective?

Reply: To answer the reviewer's query, we performed the scavenging test with the nanofibrous sheets (*Tau*-NFS & *Acr*-NFS) with a smaller area size (0, 2, 4, 6, 8, and 10 cm²). The scavenging of free DNA and free Hb has been quantified in **Fig. R6**. The scavenging efficiency increases as a function of area size to 8 cm²; after that increasing the area did not increase the scavenging efficiency. Therefore, nanofibrous sheet area size of 8 cm² is the minimum size for efficient scavenging of DAMPs. Similarly, 5 min is the shortest time for the blood to cover the scaffold area completely and is effective in scavenging the maximum amount of DAMPs.

Fig. R6. Scavenging of DNA and Hb from 42 days stored oldRBCs using different area size of nanofibrous sheets. Efficiency of nanofibrous sheets to scavenge DNA (a) and free Hb (b) increases area size of up to 8 cm², further it plateaus. Data are mean ± s.d. (n = 3); P values were determined by ordinary one-way ANOVA with Tukey’s post hoc analysis using GraphPad PRISM 9, and exact P values are indicated, ns = not significant.

Comment R3-12: P5, 2nd paragraph. Human RBC samples were treated on either day 21 or day 28 based on the rationale that this was the point at which the level of DAMPs increased exponentially (Fig 2a-f). Nevertheless, in the ‘real world’ the median storage duration of RBC concentrates is typically <21 days as reported in several large multicentre ‘age of blood’

randomized clinical trials, such as ABLE, RECESS, TRANSFUSE. It would be relevant for the authors to add comment in the Discussion.

Reply: We thank the reviewer for this suggestion. As suggested, we have added the comment in the Discussion in the revised manuscript.

Comment R3-13: P5 3rd paragraph. RBC morphology.

1) Early stages of echinocytic shape change during *in vitro* storage are reversible once the RBCs are transfused. Only counting the number of discocytes would be an under-estimate of the proportion of stored RBCs that are likely to be functional *in vivo*. The SEM images in Fig 4b show the presence of echinocyte I and II shapes in all three samples of day 42 stored RBCs (untreated, 21 days treated and 28 day treated). The authors could review their morphology counts to take this into account.

2) Did the authors assess the morphology of the RBCs immediately before and sooner after treatment, e.g., 24 hrs afterwards? It would be interesting to know whether the morphology is significantly altered by the treatment.

Reply: We agree with the argument given by the reviewer that stored old RBC units comprise predominantly cells in the early stages of echinocytosis, which may recover their shape post-transfusion, and a sizable fraction of irreparably damaged late-stage echinocytes and spherocytes (Piety NZ et al, *Transfusion* 2016;56:844-851). Hence, counting only discocytes might not reveal the potential of this technology. Typically, sphero-echinocytes are irreversibly damaged RBCs that do not recover post-transfusion, *in vivo*. Therefore, in the same samples, we counted the number of sphero-echinocytes present on 42nd day without treatment and intermittent treatment with *Tau-AcrNFS* on either the 21st or 28th day (**Fig. R7**). Data in **Fig. R7** suggest that the number of sphero-echinocytes significantly reduced in intermittently treated samples compared to the untreated stored old RBCs. Therefore, the same figure has been included in revised **Fig.4** in the revised manuscript.

To answer the second point; as the reviewer suggested, the morphology of fresh RBCs, soon after the treatment with *Tau-AcrNFS* (0.5 hours and 24 hr post-treatment) has been investigated in SEM. The SEM images in **Fig. R8** reveal that there is no effect of *Tau-AcrNFS* treatment on the morphology of RBCs.

Fig. R7. Intermittent scavenging of DAMPs using *Tau-AcrNFS* prevents structural and morphological damage to stored human RBCs.

The relative number of sphero-echinocytes present on the 42nd day revealed that intermittent scavenging of DAMPs using *Tau-AcrNFS* has decreased to 30% and 15% for the 21st and 28th day treated RBCs, respectively ($n = 4$). The number was calculated from 10,000 RBCs in each sample. Data are mean \pm s.d. ($n = 4$); P values were determined by ordinary one-way ANOVA with Tukey's post hoc analysis using GraphPad PRISM 9, and exact P values are indicated.

Fig. R8. Effect of *Tau-AcrNFS* treatment on morphology of fresh RBCs. SEM images of fresh RBCs that were untreated, and 0.5 and 24 hours post-treatment with *Tau-AcrNFS*. Scale bar = 10 μm .

Comment R3-14: P6, last paragraph. The FDA's requirement that >75% of transfused RBCs must be recoverable in the in vivo circulation 24 hrs after transfusion applies to human RBCs only. This point should be clarified here and elsewhere in the manuscript. To make comparisons with the recovery of transfused mouse RBC is irrelevant.

Reply: We agree with the reviewer that FDA's requirement only implies to humans. Nevertheless, similar standards were adopted to evaluate the transfusion efficacy in mice. (Gilson C. R. et al, *Transfusion* 2009;49:1546-1553). We observed that >75% of transfused mouse RBCs were in circulation after 24 hours of transfusion in mice. However, as the reviewer suggested, we have revised the text in the revised manuscript to clarify this point.

Comment 15: P8, 2nd paragraph, last sentence. Statement: "Therefore, three days of enhanced shelf-life in murine RBCs is equivalent to nine days in human RBCs." There is no robust science to support this extrapolation. The statement needs to be revised or deleted.

Reply: As per reviewer's suggestion, the statement is revised in the revised manuscript.

Comment 16: P8, 3rd paragraph. Mouse cytokine levels. 1) The R&D Systems mouse cytokine array tests for 40 different cytokines, chemokines, and acute phase proteins. The results for 4 bioactive factors are shown in Fig 6h. The authors should include comment about the other factors, i.e., were no differences found? 2) Ideally, images of representative arrays should be provided – as supplementary information. 3) Mouse blood samples were collected 2 hrs after transfusion (Methods P17). The authors should make this clear in the Results section, otherwise the reader would assume it was post-24 hrs.

Reply: We thank the reviewer for this suggestion. We agree with the reviewer that the Mouse Cytokine Array test can detect up to 40 different pro-inflammatory cytokines. In our studies, we detected 10 cytokines in blood plasma, out of which four cytokines data was significant, and the same has been plotted. As the reviewer suggested, the images of representative cytokine arrays are provided in the revised Supplementary Information as **Supplementary Figure 17**.

In the result section (Page 8, Para 3), it has been mentioned that blood was collected two hours post-transfusion, and cytokines/chemokines in plasma were quantified using Mouse Cytokine Array Panel A to avoid any confusion about the blood collection time point.

Comment R3-17: Discussion P9, 2nd paragraph, sentence: “When Tau-AcrNFS treatment is used terminally,.....”. By “terminally”, do the authors mean at the end of RBC storage shelf-life? Clarification is needed. P9, 2nd last line. Instead of “phenomenon”, suggest change to “hypothesis” or “expectation”.

Reply: We apologize to the reviewer for the lack of clarity. Yes, in our statement, word “terminally” means the end of RBCs storage shelf life, i.e., cleaning 42 days old RBCs with *Tau-AcrNFS*. Additionally, as reviewer suggested “phenomenon” word has been changed to “observation” which will fit better in the context.

Comment R3-18: P10, 3rd line. Author statement: “Systemic inflammation is one of the hallmarks of transfusion of stored oldRBCs”. This may be the case for transfusion of non-leukoreduced stored RBCs, but does not hold up for pre-storage leukoreduced stored RBCs. Furthermore, immunosuppression, rather than inflammation, has been documented in some clinical settings. Instead of “systemic inflammation” the authors would be advised to use the term “immunomodulation”.

Reply: As per reviewer suggestion, we have modified the text in the revised manuscript.

Comment R3-19: Methods: P13. Hemolysis assay. The equation used to the calculate the % hemolysis used a ratio of the OD readings of the sample supernatant and the 100% lysis control. This is not valid as it assumes a linear relationship between the free Hb concentration and the OD reading, which is not necessarily true. The authors need to revisit these data.

Reply: We thank the reviewer for pointing out our oversight.

In the manuscript, by mistake we gave a wrong formula as shown below, where subtraction of saline/blank values from the OD of blood supernatant was missing.

$$\frac{\text{Average OD of Blood Supernatant}}{\text{Average OD of 100\% Hemolysis control}} \times 100$$

However, we have used the blank correction in our calculations and by mistake we gave a wrong formula in the previous version. Please see the below for the correct formula that was used for calculating the % of hemolysis, which is standard in the field (Kazemi S et al Int. Aquat. Res. 2016;8:299-308). The formula has been corrected in the revised manuscript.

$$\frac{\text{Average OD of NFS treated RBCs} - \text{Average OD of saline}}{(\text{Average OD of 100\% Hemolysis control} - \text{Average OD of saline})} \times 100$$

Comment R3-20: P19. At what time-point post transfusion were the splenectomies performed on the mice?

Reply: We have mentioned the time point in the manuscript, which is 2 hrs post transfusion.

Comment R3-21: “Intermittent treatment” gives the impression of randomness, whereas the Tau-AcrNFS treatments were performed at selected time points during storage of the RBC samples. An alternative term may be better.

Reply: Although ‘intermittent treatment’ does not specify the day of the treatment, including specific days in the title becomes longer, Therefore, at present we are keeping the intermittent treatment, but throughout the manuscript wherever it is mentioned intermittent treatment, specific days of the treatment is given. We hope this will avoid any possible confusion.

Comment R3-22: Use of the verb “prevented” needs to be reviewed throughout. In most instances, the effect was not prevented per se, but rather diminished or lessened.

Reply: We thank reviewer for this suggestion. As suggested, prevented has been changed in the revised manuscript throughout.

Comment R3-23: Some of the wording could be tightened. There are instances throughout where sentences do not make sense and need to be rephrased. For example, P7, Section titled ‘Intermittent scavenging of storage lesion with Tau-AcrNFS enhances the shelf-life of stored RBCs’ has several sentences that need to be rephrased.

Reply: As reviewer suggested, we have carefully gone through the manuscript, and sentences were corrected to present with more clarity.

Comment R3-24: Statistical analysis of time course experiments in which the same sample is tested at multiple different time-points, such as for the data in Fig 2a-f and Fig 5a-b, should use repeated measures ANOVA, not one-way ANOVA with Tukey’s post-hoc.

Reply: We apologize for this oversight. We thank the reviewer for pointing this out. We now carried out statistical analysis for time course experiments using Repeated Measures ANOVA, and the revised figures have been included in the revised manuscript. Accordingly, we have modified the figure legends/Methods.

Reviewer # 4: We thank the reviewer for finding the results are interesting. Please see below for our response to each point that has been raised by the referee.

Comment R4-1: Was the degree of charges created optimized, or amounts added to functionalize?

Reply: Yes, these were systematically optimized. We have also tried a higher percentage of conjugation of charged moieties, which either formed gels in DMF or posed inadequate solubility (the data is not shown). The degree of charge plays a key role in formatting robust electrospinning nanofibrous sheets. The composition reported in the manuscript was found to be appropriate for preparing electrospinning nanofibrous sheets.

Comment R4-2: What is the loading capacity of scavenged contents per unit fiber surface area? How do you know the fibres are scavenging the components described? How much fouling of charges occurs with other components such as plasma proteins, decreasing potential efficiency? Is this "other" absorption key or detriment?

Reply: We thank the reviewer for these critical points. Please see the explanation for those points below.

Our data suggest that DAMPs not physically adsorbed onto the fibers, instead they are scavenged through electrostatic interactions and intercalation. For example, the designing of the cationic and anionic polymers is the critical parameter to show the scavenging of DAMPs. To capture positively charged free histones in blood units, we have developed an anionic nano-fiber scaffold by using poly-aurine. The choice of aurine is based on the mechanism performed in the body where circulating free histones and chromatin clearance are mediated by charge interactions with cell surface HSPG (Heparan sulphate proteoglycan) through its sulphate moiety (Clos D. et al, *Clinical & Experimental Immunology* 1999;117:403-411). Additionally, aurine could also bind to free Hb.

On the other hand, nanofibrous sheets were systematically designed to bind efficiently to extracellular DNA not only through electrostatic interactions but through intercalation as well. A combination of electrostatic interactions and intercalation enhances the DNA scavenging capacity. For example, we have prepared four nanofibrous sheets with four different polymers, poly-tetradecane, poly-pyridinium, poly-YO, and poly-acridine (**Fig. R3a**). While poly-tetradecane is a neutral polymer, poly-pyridinium, poly-YO and poly-acridine have a cationic charge. However, poly-pyridinium does not intercalate with DNA. On the contrary, YO and acridine are DNA intercalators. However, acridine is a stronger intercalator than YO (Kulyk O. G. et al, *Dyes and Pigments* 2022;200:110148 and Larsson A. et al, *J. Am. Chem. Soc.* 1994;116:8459-8465). Furthermore, we have prepared nanofibrous sheets using these polymers, subsequently DNA scavenging studies were carried out with 42 days stored old RBCs. The relative DNA scavenging efficiency has shown in **Fig. R3b**. The data in **Fig. R3b** shows that scavenging efficacy is in the following order, *Tetradecane*-NFS < *Pyr*-NFS < *YO*-NFS < *Acr*-NFS.

These results suggest that electrostatic interactions and intercalation play a key role in DNA scavenging.

Fig. R3. Ionic interactions and intercalation enhance DNA scavenging efficiency. a, Chemical structures of synthesized polymers that were used to generate nanofibrous scaffolds. **b,** Relative DNA presence in stored old RBCs before scavenging and after scavenging with various nanofibrous scaffolds. For **b,** *P* values were determined by ordinary one-way ANOVA with Tukey's post hoc analysis using GraphPad PRISM 9,

As the reviewer pointed out, the fouling of charge could occur with proteins. In the event of fouling of charge, the overall efficiency of charged nanofiber sheets will get affected. However, the maximum amount of proteins remain in the plasma. During the packing of stored RBCs, typically, plasma is removed. Therefore, all plasma proteins are removed. Hence, plasma proteins mediated fouling of charge would not occur. However, *in situ* generated proteins could bind to nanofibers during the storage of RBCs to reduce the efficiency of the DAMPs scavenging process. To evaluate the fouling of charge by RBCs/WBCs protein from packed RBCs unit, positively charged *Acr*-NFS were incubated with a negatively charged protein BSA (Bovine Serum Albumin, 0.2%) for one hour. Subsequently, DNA from stored oldRBCs was scavenged using either native *Acr*-NFS or BSA-incubated *Acr*-NFS. Data in **Fig. R9a** suggests that the scavenging efficiency of *Acr*-NFS significantly reduced the fouling of charges with BSA protein. Similarly, negatively charged *Tau*-NFS were incubated with a positively charged protein holo-Transferrin bovine (0.2%) for one hour. Subsequently, free Hb was scavenged using either native *Tau*-NFS or transferrin incubated *Tau*-NFS. Data in **Fig. R9b** suggests that the scavenging efficiency of *Tau*-NFS significantly reduced the fouling of charges with transferrin protein. Cumulatively, these results suggest that excessing fouling of charges could reduce the efficacy of nanofibers to efficiently scavenge DAMPs. Therefore, it is important to use these NFS without plasma samples.

Fig. R9. Masking of ionic charges decreases the DAMPs scavenging efficiency of nanofibrous sheets. **a**, Scavenging of DNA using either native *Acr*-NFS or *Acr*-NFS preincubated with negatively charged BSA. **b**, Scavenging of Hb using either native *Tau*-NFS or *Tau*-NFS preincubated with positively charged Transferrin. *P* values were determined by ordinary one-way ANOVA with Tukey's post hoc analysis using GraphPad PRISM 9, and exact *P* values are indicated.

Comment R4-3: In the introduction the authors state that "technologies to prevent the formation of DAMPs in stored RBCs remains an unmet need." This proposed technology is not prevented DAMPs but for chelating them to prevent associated damage.

Reply: We thank the reviewer for suggesting the modification. We have incorporated the change as per reviewer's suggestion.

Comment R4-4: More studies are needed to show above and that scavenging is occurring and determining the loading capacity.

Reply: These points were systematically answered with additional data, please see the reply for previous comments.

Once again, we thank the editor and all reviewers for their time for critically reading the manuscript and providing their valuable suggestions, which have significantly strengthened the manuscript.

Based on our revision of the manuscript in response to the reviewers' constructive feedback, we believe the revised manuscript is now suitable for publication in your esteemed journal, *Nature Communications*.

REVIEWER COMMENTS

Reviewer #1 (Remarks to the Author):

The authors have adequately addressed all questions and comments in the revision. In my view, the manuscript is ready for publication in Nature Comm.

Reviewer #2 (Remarks to the Author):

The suggestions have been included in the revised manuscript satisfactorily.

Reviewer #3 (Remarks to the Author):

This reviewer (previously identified as Reviewer #3) acknowledges the genuine efforts of the authors to address this reviewer's comments, which mostly are satisfactory. However, some queries remain – identified below, using the numbering code in the authors' Rebuttal.

R3-1. In their reply regarding the issue of leukoreduction filtration, the authors note that an important driver for their research was the desire to develop an inexpensive intervention that could be used in developing and under-resourced countries where non-leukoreduced blood products are the standard inventory. This is a very valid point and should be included in the Introduction.

R3-2. Suggest rephrase that last part of the new sentence (Discussion, bottom of pg10, starting "We are designing Tau-AcrNFS based insert in blood bags as a medical device, ") to read "....., which will be feasible to use in the blood bank processing setting without the need for multiple interventions during storage of RBCs."

R3-3. This reviewer is still not entirely convinced by the authors' reply regarding the claimed "novelty" of the scavenging chemistries of their Tau-AcrNFS technology (i.e. electrostatic interactions and intercalation). To the extent of this reviewer's understanding, the data shown in Fig R3b in the Rebuttal does not really prove anything about the actual nature of the chemical interactions involved. It would be prudent for the authors to soften their claims.

R3-4. Hemocompatibility. This reviewer acknowledges the additional experiments performed by the authors, however simply quantifying the numbers of cells and hemolysis following exposure to the Tau-AcrNFS treatment is insufficient. Ultimately, discriminative and highly sensitive chemical analyses, such as by untargeted LC-MS/MS, will be required to determine any biochemical changes following exposure of blood to Tau-AcrNFS treatments. The authors need to acknowledge this in the manuscript.

R3-5. In their reply, the authors have provided clearer description about how the blood samples were treated with the Tau-AcrNFS, but these additional details do not seem to have been included in the Methods sub-section 'Scavenging of DAMPs from stored human RBCs using Tau-NFS and Acr-NFS, in vitro'. Please review.

R3-10 (first point). This reviewer could not identify the new sentence in the revised manuscript that the authors mention in their reply. Please specify where this new sentence is located.

R3-10 (third point). This reviewer remains puzzled by the hemolysis criteria (i.e. <5%) used by the authors. How does the ASTM F756-17 standard referred to by the authors in their reply relate to the FDA regulation for stored RBCs, which stipulates <1% hemolysis (the Council of Europe guidelines specify <0.8% hemolysis)? This will be a source of confusion for readers from the transfusion/blood

banking fields. The authors need to review.

R3-19. Hemolysis calculation. It is incorrect to use the actual OD readings in the calculation of % hemolysis. The reported % hemolysis results will be inaccurate. The correct way to do the experiment is to prepare a 100% lysis sample of the untreated RBC specimen and then prepare a series of dilutions to generate a standard curve. The level of hemolysis of the Tau-AcrNFS-treated RBCs is then derived from the standard curve. The authors need to review.

Reviewer #4 (Remarks to the Author):

The authors have done a great job at responding to original review comments/suggestions.

We thank all the reviewers for their positive response, critical feedback and recommending for publication. As few minor points were raised, we have answered those queries and revised the manuscript accordingly. Please see below for our response to each point that the reviewer has raised.

Reviewers # 1, 2 and 4. We glad to know that reviewers are satisfied with the revised manuscript, and we thank the reviewers for recommending for publication.

Reviewer # 3. We thank the reviewer for critical feedback and appreciating our efforts to address the queries raised during the first revision. Please see below for our response to each point that has been raised by the referee.

Comment R3-1: In their reply regarding the issue of leukoreduction filtration, the authors note that an important driver for their research was the desire to develop an inexpensive intervention that could be used in developing and under-resourced countries where non-leukoreduced blood products are the standard inventory. This is a very valid point and should be included in the Introduction.

Reply: We thank the reviewer for this suggestion. As suggested, we have included this point in the revised introduction. (Page 3, Text in blue color)

Comment R3-2: Suggest rephrase that last part of the new sentence (Discussion, bottom of pg10, starting “We are designing Tau-AcrNFS based insert in blood bags as a medical device, “) to read “....., which will be feasible to use in the blood bank processing setting without the need for multiple interventions during storage of RBCs.”

Reply: As the reviewer suggested, we have modified the sentence.

Comment R3-3: This reviewer is still not entirely convinced by the authors’ reply regarding the claimed “novelty” of the scavenging chemistries of their Tau-AcrNFS technology (i.e. electrostatic interactions and intercalation). To the extent of this reviewer’s understanding, the data shown in Fig R3b in the Rebuttal does not really prove anything about the actual nature of the chemical interactions involved. It would be prudent for the authors to soften their claims.

Reply: As the reviewer suggested, throughout the manuscript we have soften our claims about chemical interactions.

Comment R3-4: Hemocompatibility. This reviewer acknowledges the additional experiments performed by the authors, however simply quantifying the numbers of cells and hemolysis following exposure to the Tau-AcrNFS treatment is insufficient. Ultimately, discriminative and highly sensitive chemical analyses, such as by untargeted LC-MS/MS, will be required to

determine any biochemical changes following exposure of blood to Tau-AcrNFS treatments. The authors need to acknowledge this in the manuscript.

Reply: We completely agree with the reviewer's point. Quantification of cell number and hemolysis data provides a partial information on hemocompatibility. A quantitative hemocompatibility can be measured as the reviewer suggested. For example, Garry Corthal's group has done a significant work in mapping the erythrocyte membrane proteins (Kottahachchi D, et al. *EuPA OPEN PROTEOMICS* 2015, 7, 43-53). However, such a detailed LC-MS/MS study is not feasible for us within in this time frame, therefore, we have removed the hemocompatibility claim, and modified the text in the revised manuscript (including in Figure legends), and added a note that a quantitative hemocompatibility study is needed and will be done in the next study.

Statement like "to measure the hemocompatibility" was removed and modified as "to understand whether treatment of *Tau*-NFS and *Acr*-NFS can cause cell loss"....

Comment R3-5: In their reply, the authors have provided clearer description about how the blood samples were treated with the Tau-AcrNFS, but these additional details do not seem to have been included in the Methods sub-section 'Scavenging of DAMPs from stored human RBCs using Tau-NFS and Acr-NFS, in vitro'. Please review.

Reply: We apologize for missing this point. As the reviewer suggested, now we have included these details in the Methods section.

Comment R3-10 (first point): This reviewer could not identify the new sentence in the revised manuscript that the authors mention in their reply. Please specify where this new sentence is located.

Reply: We thank the reviewer for pointing this out. We have added this sentence, please see Page-5, end of third para. Highlighted the text in blue colour.

Comment R3-10 (third point): This reviewer remains puzzled by the hemolysis criteria (i.e. <5%) used by the authors. How does the ASTM F756-17 standard referred to by the authors in their reply relate to the FDA regulation for stored RBCs, which stipulates <1% hemolysis (the Council of Europe guidelines specify <0.8% hemolysis)? This will be a source of confusion for readers from the transfusion/blood banking fields. The authors need to review.

Reply: As the reviewer suggested, we have removed claims of accepted range of hemolysis is <5% to avoid any confusion for readers. Instead, we have stated that upon treatment with *Tau-AcrNFS* <1% hemolysis was observed (**Fig. 1f**).

Comment R3-19: Hemolysis calculation. It is incorrect to use the actual OD readings in the calculation of % hemolysis. The reported % hemolysis results will be inaccurate. The correct way to do the experiment is to prepare a 100% lysis sample of the untreated RBC specimen and then prepare a series of dilutions to generate a standard curve. The level of hemolysis of

the Tau-AcrNFS-treated RBCs is then derived from the standard curve. The authors need to review.

Reply: We sincerely thank the reviewer pointing this mistake and suggesting the best way of conducting hemolysis assay. As the reviewer suggested, we have redone hemolysis assay. We have prepared a 100% lysis of untreated RBCs through exposing to Triton-X, and measured the Hb concentration through Drabkin's assay, which was considered as 100%. Subsequently, a series of dilutions were made and a standard curve was generated. Then the level of hemolysis of *Tau-AcrNFS* treated RBCs sample was derived from the standard curve. In this method, the observed hemolysis was 0.93% (<1%). Accordingly, we have replaced the previous **Fig. 1f** panel with the recent data, and revised the text in the figure legend for **Fig. 1f**.

Once again, we thank the reviewers for their time for providing their valuable suggestions. Based on our revision of the manuscript in response to the reviewers' constructive feedback, we believe the revised manuscript has strengthened.

REVIEWERS' COMMENTS

Reviewer #3 (Remarks to the Author):

In this further revised manuscript, the authors have satisfactorily addressed all of this reviewer's comments (Reviewer #3).

No further comments.